# Triacylglycerol Composition and Chemical-Physical Properties of Cocoa Butter and Its Derivatives: NMR, DSC, X-ray, Rheological Investigation

**DOI:** 10.3390/ijms24032090

**Published:** 2023-01-20

**Authors:** Maria Francesca Colella, Nadia Marino, Cesare Oliviero Rossi, Lucia Seta, Paolino Caputo, Giuseppina De Luca

**Affiliations:** 1Department of Chemistry and Chemical Technologies (CTC), University of Calabria—UNICAL, Via P. Bucci, Arcavacata di Rende, 87036 Rende, Italy; 2Reolì S.r.l., Zona Industriale, Settore 3, 87064 Corigliano-Rossano, Italy

**Keywords:** cocoa butter (CB), cocoa butter derivatives, Differential Scanning Calorimetry (DSC), NMR spectroscopy, rheology, powder X-ray (PXRD)

## Abstract

In recent years, the food industry has become increasingly involved in researching vegetable fats and oils with appropriate mechanical properties (ease of transport, processing, and storage) and a specific lipidic composition to ensure healthy products for consumers. The chemical–physical behavior of these matrices depends on their composition in terms of single fatty acids (FA). However, as we demonstrate in this work, these properties, as well as the absorption, digestion and uptake in humans of specific FAs, are also largely determined by their regiosomerism within the TriAcylGlycerols (TAG) moieties (*sn*-1,2,3 positions). The goal of this work is to study for the first time vegetable fats obtained directly from a sample of natural cocoa butter (CB) through a process that manipulates the distribution of FAs but not their nature. Even if the initial percentage of each FA in the mixture remains the same, CB derivatives seem to show improved chemical–physical features. In order to understand which factors account for their physical and chemical characteristics, and to check whether or not the obtained new matrices could be considered as valid alternatives to other vegetable fats (e.g., palm oil (PO)), we carried out an experimental investigation at both the macroscopic and molecular level including: (i) Differential Scanning Calorimetry (DSC) analyses to examine thermal features; (ii) rheological testing to explore mechanical properties; (iii) powder X-ray diffraction (PXRD) to evaluate the solid-state phases of the obtained fats; and (iv) ^1^H and ^13^C Nuclear Magnetic Resonance (NMR, 1D and 2D) spectroscopy to rapidly analyze fatty acid composition including regioisomeric distribution on the glycerol backbone. These last results open up the possibility of using NMR spectroscopy as an alternative to the chromatographic techniques routinely employed for the investigation of similar matrices.

## 1. Introduction

Lipids are among the most abundant compounds in nature. They are generally found in the form of triglycerides, also called TriAcylGlycerols (TAGs), in which different fatty acids (FAs) are esterified to the glycerol backbone (Figure 1a) [1,2]. FAs are mainly hydrocarbon biomolecules, with an even number of carbon atoms and a carboxylic functional group at one end of the chain. According to the number of π-bonds found in the acyl chain, it is possible to distinguish saturated (SFAs), mono-unsaturated or monoenoic (MUFAs), and polyunsaturated (PUFAs) fatty acids [2].

Fats are also the most widely used raw material in various industrial fields, principally in the food sector. Palm oil (PO) is the most employed fat, especially in the production of pastry, because it possesses all the features that the confectionary requires, e.g., easy transportation and storage, and great machinability [3,4]. However, PO is the subject of a highly controversial issue that goes beyond simple health questions. Indeed, despite recent attempts have to make its cultivation sustainable, the PO market is considered one of the greatest threats to tropical biodiversity [5,6,7] and the cause of adverse environmental impacts, such as habitat loss, massive carbon emissions, pollution of water and air, deforestation, and forest fires [8,9,10].

As a consequence, in food industry research, ease of transport, processing, and high melting temperature of products must reconcile with the consumers’ request to use vegetable alternatives to PO with equivalent features and with a beneficial impact on human health [11,12,13,14]. The biggest problem is that most of the valid alternative fats, such as avocado, olive, sunflower, or dried fruit oils, while having a very good ratio of unsaturated to saturated fatty acids, are liquid at room temperature due to their chemical composition [15]: carbon chains of FAs in TAGs are principally mono-or poly-unsaturated with a double bond in a *cis*-configuration (meaning with a rigid structure), and so cannot pack together closely enough to solidify. Generally, different oils and fats show specific chemical and physical behavior largely determined not only by the type and quantity of FAs, but also by the stereospecific positioning (*sn*-; i.e., stereospecific numbering) of FAs on the glycerol backbone [2,16]. For example, the presence of an enoic chain exclusively on the terminal position (*sn*-1,3) and a MUFA like oleic acid at the *sn*-2 position determines that exotic fats like palm oil (PO) or cocoa butter (CB) are solid or semi-solid at room temperature [17]. Moreover, the stereospecific structure of TAGs, together with the acyl chain length, is also linked to a specific, improved metabolism and absorption of lipids in the human body [16,18,19,20]. That is why industrial research has a great interest in understanding how the chemical composition of these fats can influence their chemical and physical behavior. For example, CB is an interesting subject matter in food science and an essential ingredient in the baking industry: it is responsible for the texture, gloss, and typical melting behavior of chocolate [21,22]. It is solid at room temperature because it is largely composed (over 70%) of symmetrical triglycerides with an unsaturated fatty acid (generally indicated by the letter “U”) at the *sn*-2 position. These include principally oleic acid (O, C18:1), and linoleic acid (L, C18:2) at a much lower percentage. Saturated FAs (indicated by the letter “S” and mostly palmitic (P, C16:0) and stearic (S, C18:0)) are, instead, in the *sn*-1 and *sn*-3 positions of the glycerol backbone. This type of TAG is generally indicated as SUS (Saturated-unsaturated-saturated). Going into the details of the chemical composition of CB in terms of TAGs, POS (1-palmitoyl-2-oleoyl-3-stearyl-sn-glycerol; 37–39%; Figure 1b), SOS (1,3-stearyl-2-oleoyl-sn-glycerol; about 25%) and POP (1,3-palmitoyl-2-oleoyl-sn-glycerol; about 15%) are mainly present. These percentages may vary slightly depending on the geographical origin of CB and this causes moderate variations in its physical properties such as moderate variations in melting temperatures [23,24,25].

As mentioned previously, CB is a solid fat at room temperature (20–25 °C) and melts completely at body temperature [22,23]. This low melting temperature is the reason for its use as the main ingredient in the production of chocolate, whose peculiar feature is to melt as soon as it is in the mouth. At the same time, this low melting temperature, coupled with the fact that it forms grains over time in blends, makes it difficult to use CB in other industrial food products [22,23,24,25,26]. On the other hand, the food market is strongly interested in exploring new processes aimed at modifying these vegetable materials to obtain more available and easily workable alternatives, which would also be advantageous for both the market and human health [17]. In this regard, some manufacturers are trying to find alternative methods to chemical and enzymatic routes for modifying the structure of TAGs in CB to expand its applications [27,28,29]. Thus, in this context, the present study was designed to experimentally investigate the changes in the chemical–physical properties of some modified fats derived from a natural CB sample through an unconventional process. The aim is to demonstrate that this modification process has led to a manipulation of the positional distribution of the FAs, but has left their nature unaltered, i.e., following the modification process, positional isomers have been obtained in which part of the FAs in the mixture of SUS-type TAGs (mainly POP, POS and SOS) were converted into SSU-type (Saturated-saturated-unsaturated-type), mainly PPO (1,2-palmitoyl-3-oleoyl-sn-glycerol), PSO (1-palmitoyl-2-stearyl-3-Oleoyl-sn-glycerol), SPO (1-stearyl-2-palmitoyl-3-oleoyl-sn-glycerol) and SSO (1,2-stearyl-3-oleoyl-sn-glycerol).

To this end, we carried out a comparative study among the native CB sample, three samples of fats derived from it, and a sample of PO. The TAGs composition, crystallization, thermal behavior, and rheological properties have been studied on all these materials in order to understand their characteristics, properties, and differences at the molecular and macroscopic level. In particular, the experimental investigation included: (i) Differential Scanning Calorimetry (DSC) tests to examine thermal features (mainly melting points); (ii) rheological testing to explore mechanical properties; (iii) powder X-ray diffraction (PXRD) to qualitatively compare the obtained matrices at the solid-state level; and (iv) ^1^H and ^13^C Nuclear Magnetic Resonance (NMR, 1D, and 2D) spectroscopy for molecular characterization. It should be emphasized that the NMR technique was particularly useful in the qualitative and quantitative (in the percentage of TAGs) characterization of these complex mixtures and, it allowed us to experimentally confirm the positional rearrangement of the acyl chains that occurred during the modification process.

The accordance between experimental results from these different techniques confirms that a specific chemical and physical behavior is strongly determined by the stereochemistry of TAGs.

## 2. Results and Discussions

### 2.1. Differential Scanning Calorimetry

The thermal behavior of both natural CB and cocoa butter equivalent (CBE) has been extensively investigated using DSC methods in order to study the different crystalline forms and the melting behavior they may present, as demonstrated by the plethora of papers present in the literature [30,31,32,33,34,35,36]. In the present study, the DSC melting profile of the native CB and of the three modified samples (sample B, C and D) was used to make a first rapid assessment of their melting temperature as a function of the modification process.

Figure 2 shows these melting profiles for all the samples analyzed. As can be seen, the melting thermogram of natural CB, sample A, shown in Figure 2a, is characterized by a rather large endothermic peak centered at about 29 °C, indicating a single-phase transition from the solid to the liquid state. Conversely, the melting thermograms of the modified samples, samples B, C, and D, shown in Figure 2b, appear more complicated. Indeed, they display a pronounced signal similar in location and shape to that of natural CB, with one or two large peaks shifted to higher temperatures. Additionally, the calorimetric curves of all samples showed only endothermic peaks. These data clearly indicate that the modified butter samples present a much more complicated polymorphic system, with high transition temperatures (up to about 52.6 °C). This is difficult to explain because of the presence of different crystalline forms due to the unique TAG mixture present in the natural butter (i.e., SUS-type, mainly POP, POS, and SOS). To attempt to rationalize this effect we hypothesized that, as will be confirmed later by the molecular characterization via NMR, the modification reaction may lead to the formation of SSU-type TAGs (i.e., Saturated-saturated-unsaturated type, mainly PPO, PSO, SPO, SSO) which coexist with the SUS-type. In this case mixtures of TAGs with different stereochemistry in different ratios could have a strong effect on the melting point. This hypothesis seems to be confirmed also by the thermal behavior of the modified sample D, whose thermogram, reported in Figure 3, is compared with that obtained under the same conditions for PO. As can be seen from this comparison, the melting thermograms of the two samples are nearly identical with only minor differences in the melting temperatures. The melting peaks in the PO thermogram have been attributed to the presence of particular triacylglycerols as a function of their composition and the regioisomeric distribution of the acyl chains, as demonstrated by previous studies [37]. Moreover, the melting enthalpy (ΔH*_m_*) of all the samples was calculated and the results are reported in Table 1.

### 2.2. Rheological Measurements

In Figure 4, the time cure curve of native CB and modified samples, B, C, and D are reported. Figure 4a shows the time cure experiments for native CB whose mechanical behavior is that of a liquid-like Newtonian system and shows a clear phase transition, indicated by a sharp change in slope for both moduli (G′ and G″), from solid to liquid around 30 °C. Around this temperature, a crossover is observed between the two moduli until the complete disappearance of the elastic modulus at around 34 °C. Conversely, the time cure tests of modified samples are more complex. Indeed, for sample C (Figure 4c) it is still possible to observe a single transition from solid to liquid, but this occurs gradually, with a smooth change in the slope of the two moduli, and in the temperature range from 23 to 40 °C. Samples B and D (Figure 4b,d, respectively), show a similar profile with two different phase transitions (solid to viscoelastic and viscoelastic to liquid) but with distinct temperature ranges. Sample B is solid until 29 °C, when its behavior is viscoelastic in the temperature range of 29–35 °C, and finally, it becomes liquid at 39 °C. Sample D is solid until around 36 °C, then it becomes viscoelastic between 36 °C and 45 °C, and becomes liquid at around 50 °C. These experimental results are in line with those obtained from the DSC experiments and corroborate the advanced hypothesis: namely, that although all the samples are made up of mixtures of different TAGs, the native CB, and in part also sample C, appears as a pseudo-single-component system, in which SUS-type symmetric TAGs (i.e., POP, POS, and SOS) statistically prevail. The modified samples B and D are more complicated systems with more mixtures of TAGs in which some SUS-type TAGs are converted into SSU-type (i.e., PPO, PSO, SPO, SSO), causing tighter packing and strongly increasing the transition temperature at which both samples are completely melted (up to at about 53–54 °C). Moreover, it is interesting to note that the rheological behavior of sample D is, once again, very similar to that of PO as can be seen in Figure 4d, where the comparison between their time cure curves is shown.

### 2.3. Powder X-ray Diffraction

In order to have a complete overview of the solid-state characteristics of the modified CB fats, powder X-ray diffraction (PXRD) measurements were also conducted on our samples and on native CB and PO. PXRD is the most common technique used to evaluate the crystal packing and the polymorphic state of solid and semi-solid materials, both of which can influence the rheology of the samples [30,31,38]. In this work, however, a procedure was followed that minimized the possibility of random packing, allowing for measurement reproducibility and for direct, comparative analysis of all samples, which was our goal at this stage, rather than a deep investigation of the polymorphism eventually showed by each examined fat. In particular, three heating and cooling cycles were performed on each sample before the measurement, and all the samples were manipulated in parallel, placed in the same type of holder, and measured in batteries (one after the other). The result of the qualitative investigation is summarized in Figure 5a–c, in which the PXRD profiles of native CB, modified samples B and D, and PO, are shown and selectively compared. This study confirmed that the more modified the fat with SUS-type TAGs converted into SSU-type (sample D has the highest degree of modification, as will be confirmed later by NMR data), the closer its PXRD profile is to that of PO, in accordance with calorimetric and rheological data (Figure 5c).

### 2.4. ^1^H NMR: Natural Cocoa Butter and Modified Samples

According to previous studies [30,31,32,33], we know that natural CB is mainly a mixture of SUS-type TAGs, with an unsaturated (U) and/or polyunsaturated fatty acids in *sn*-2 position and a saturated (S) fatty acid in *sn*-1 and *sn*-3 sites of the glycerol backbone.

Although this situation changed in the modified samples (B, C, and D), as will be discussed later, no significant difference emerged from the protonic profile when the ^1^H NMR spectrum of natural CB (sample A) was compared with that of modified samples (Figure 6). Therefore, for the sake of simplicity, discussion and assignment of signals will be reported only for to the natural sample.

The accordance between experimental data from the ^1^H NMR spectra, validated by the ^1^H-^1^H COSY NMR experiment, and results of previous studies, allowed us a complete characterization of the samples. Because the ^1^H NMR spectra for natural and modified samples, as expected, are very similar to those of other well-known common vegetable oils [39,40], we will not go into the details of the assignment process. All the protonic signal attributions are summarized in Table 2 and highlighted both in ^1^H (Figure 7) and ^1^H-^1^H COSY NMR (Figure 8) spectra.

### 2.5. Parallel Study of ^13^C NMR Spectra of Natural and Modified Cocoa Butter

NMR ^13^C-{^1^H} spectra of natural CB (Figure 9) and of modified samples are much more informative and interesting than proton spectra: visible substantial differences in terms of chemical shifts and intensity are present in the four spectra. Indeed, for samples B, C, and D, new signals can be seen which are able to differentiate the carbon signals in the various positions of the TAGs, in addition to those signals recognized in the case of the starting matrix. For this reason, a clearer enumeration of carbon nuclei of fatty acid chains is necessary (Figure 10): following the IUPAC system [41] the carboxyl carbon for all FAs is denoted by the number one, and positions in the chain for other carbon atoms are denoted with reference to it.

In order to simplify the discussion, and because the situation is very close to that of the other modified fats studied, only the comparison between the natural CB (sample A) and one of the modified fats (sample B) will be discussed.

Based on experimental data obtained from the 1D ^13^C-{^1^H} and 2D ^1^H-^13^C HMQC correlation experiments and confirmed by data from the literature [42,43,44], the signal attribution for the complete lipidic characterization of natural CB sample was carried out [45].

As can be seen from Table 3, for sample B in many areas of the spectrum there are two values of chemical shift for the same carbon of the various acids as a function of the acyl chain position (*sn*-1,3 or *sn*-2) in the TAG. In particular, there are three regions of the ^13^C NMR spectrum where this effect is much more noticeable: the region ranging from 33.80 ppm to 34.50 ppm, which is assigned to the C-2 carbons; the region from 127.70 ppm to 130.30 ppm assigned to the olefinic carbons of oleic and linoleic acids; and, finally, the region from 172.50 ppm to 173.60 ppm assigned to the C-1 carbonyls.

Figure 11 shows these three enlarged areas of the ^13^C spectrum in which the spectrum profile of natural CB is compared with that of modified sample B. For example, if we consider the region of carbons C-2 from 33.80 to 34.50 ppm (Figure 11a), for natural CB we can see only two signals: one at 34.08 ppm and one at 34.22 ppm assigned, respectively, to the C-2 of palmitic and stearic acids at the *sn*-1,3 position, and to C-2 of oleic and linoleic acids at the *sn*-2 position. These two peaks correlate to the H_A_ proton in the HMQC map. Instead, in the spectrum of the modified sample B, in addition to the peaks already identified, two other well-defined signals with slightly different chemical shifts are seen: the first one at a lower frequency, 34.05 ppm; and the second at a higher frequency, 34.24 ppm. These two peaks have been assigned to C-2 carbons of oleic and linoleic acids in position *sn*-1,3, and to C-2 carbons of palmitic and stearic acids in position *sn*-2, respectively. Even in this case, the confirmation of the correct assignments is given by the correlation in the HMQC map between these signals and those belonging to the proton H_A_. The appearance of these extra signals in the ^13^C spectrum of sample B compared to the spectrum of CB also occurs in the other regions of the spectrum as shown in Figure 11b,c.

This experimental evidence can be explained by assuming that in the modified sample B, some of the fatty acids in the TAG mixture, mainly of the SUS-type (i.e., POP, POS, SOS), modify their positional distribution with the formation of the SSU-type (i.e., PPO, PSO, SSO). Hence, the treatment to which the native fat was subjected only had the effect of modifying the positional distribution of the acyl chains on the TAGs, leading to a mixture of SUS- and SSU-type TAGs. The presence of these extra peaks also occurs in the ^13^C spectra of the other modified samples C and D, with the only difference from sample B being in the relative intensities of the extra peaks. This means that the SSU-type TAGs’ formation percentage is different in the three modified butters.

Additionally, by recalling that NMR spectroscopy is a quantitative technique, and since the ^13^C spectrum was recorded with the *zgig* pulse sequence, i.e., quantitative ^13^C NMR spectroscopy with inverse gated ^1^H-decoupling, the signals can be integrated to obtain the relative percentages of the modified TAG SSU-types in samples B, C and D. This is a relative quantitative determination that focuses mainly on previously observed regions, where the observed differences in the chemical shifts of carbon atoms as the chains move from *sn*-1,3 to *sn*-2 positions or vice versa are most evident. In particular, for these quantitative determinations, the signals of the carbonyl carbons (C-1) and of the α-carbons (C-2) were taken into consideration. To note that the lower limits of detection (LOD) are typically within an order of magnitude of micromolar (about 10^−6^) [38]. Since the signals are too close to each other, deconvolution was chosen as the integration method in order to establish the ratio of saturated to unsaturated fatty acids for both the *sn*-1,3 and *sn*-2 positions. The data of this integration process are reported in Table 4 for samples A, B, C, and D and they are the results of the mean of three replicates.

Moreover, looking at both ^1^H and ^13^C NMR spectra, there is no experimental evidence of the possible formation of *trans* acids during the modification process, and therefore it appears the fats did not undergo excessive degradation during the process. Studies are still in progress to investigate possible degradation reactions that may have occurred in the formation of compounds B, C, and D. We would like to underline that these results lead us to conclude that NMR spectroscopy, and in particular ^13^C NMR methodology, provides the most convenient and rapid way for the determination of the composition and regioisomeric distribution of the acyl chains in glycerol tri-esters, without requiring a pretreatment of the sample [46,47,48] as is the case for the chromatographic technique commonly used for this purpose [46,49,50].

**Figure 10 ijms-24-02090-f010:**
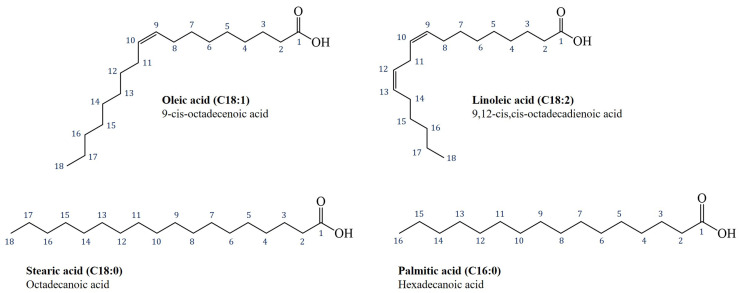
Structure and a clearer labeling (IUPAC system) for the carbon nuclei of the main and most abundant fatty acids in natural CB [51]. The carboxyl carbon for all FAs is denoted by the number one, and positions in the chain for other carbon atoms are denoted with reference to it. For simplicity, the molecules are represented as free fatty acids instead of their esterified form as TAGs.

**Figure 11 ijms-24-02090-f011:**
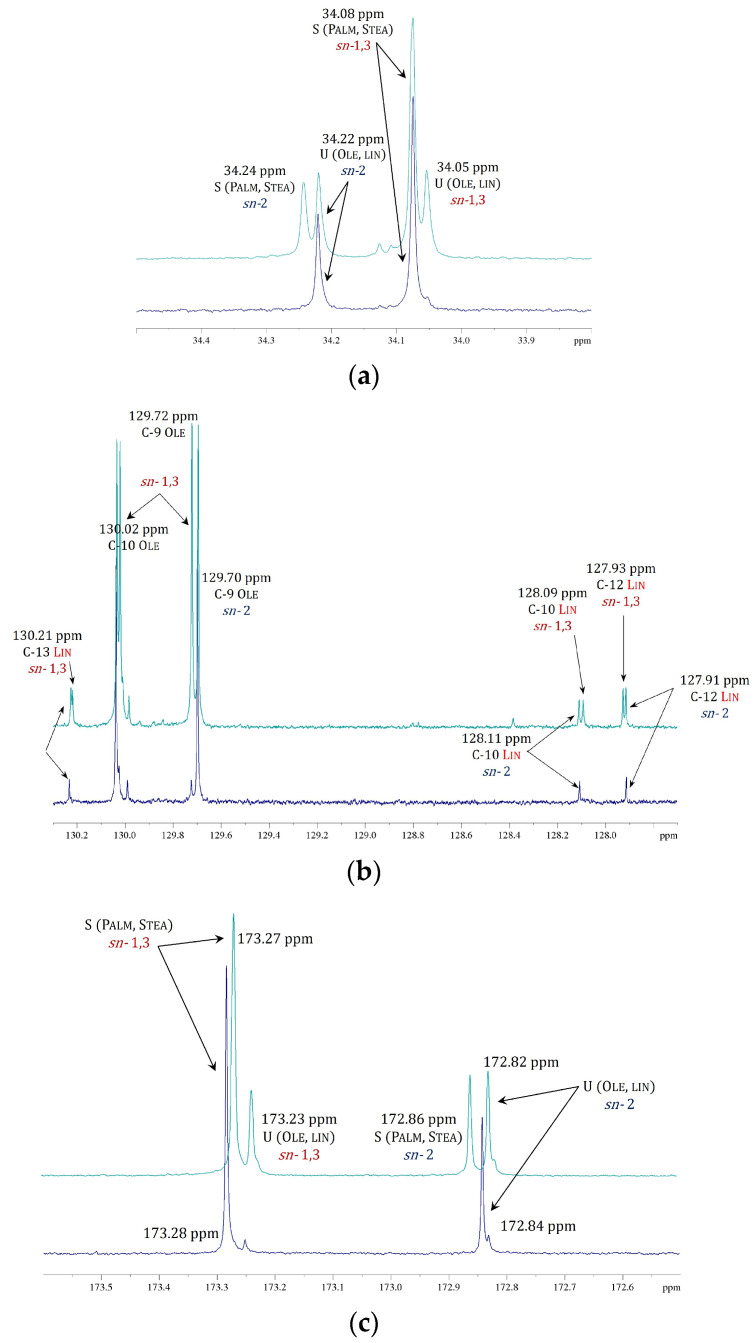
Comparison of ^13^C NMR spectra (125 MHz, 298 K, in CDCl_3_) for natural CB (sample A, in blue) and modified butter (sample B, green) in the range of (**a**) the α-carbons signals C-2 [33.80–34.50 ppm], (**b**) olefinic signals of mono-unsaturated and di-unsaturated acyl chains [127.70–130.30 ppm] and (**c**) of the carbonylic carbon signals of mono-unsaturated and saturated fatty acids [172.50–173.60 ppm].

## 3. Materials and Methods

### 3.1. Starting Materials and Modified Fats

All measurements were performed on deodorized commercial samples of CB and PO. In order to evaluate the effect of structural variation in TAGs on nutritional, organoleptic, and mechanical properties of the vegetable fats, the following was materials were used: Starting with the natural matrix (sample A), three different modified fats were obtained (samples B, C, and D) through a process carried out without the enzymes or chemical additives conventionally employed [28,29]. In this case, natural CB was treated by appropriately regulating temperature and pressure under a controlled atmosphere. The three samples B, C, and D were obtained under different experimental conditions, but since the process will be covered by an industrial patent for obvious reasons of secrecy, it is not possible to reveal the precise details on the three processes.

### 3.2. Differential Scanning Calorimetry

Differential Scanning Calorimeter (DSC) experiments to study the thermal behavior of fat samples were performed using a DSC SETARAM 131 instrument in an inert nitrogen atmosphere [32,51].

The following heating/cooling cycles were performed for all the samples:Isotherm at 15 °C for 20 minHeating from 15 °C to 70 °C at 2 °C/minCooling from 70 °C to 15 °C at 2 °C/minIsotherm at 15 °C for 10 minHeating from 15 °C to 70 °C at 2 °C/min

The preliminary steps (1–4) were necessary to avoid any thermal history effects and to ensure an identical starting condition for all the samples. Consequently, DSC profiles obtained from step 5 are reported in the figures [52].

### 3.3. Rheological Measurement

Dynamic Shear Rheometer (DSR) measurements on natural and modified CB samples were carried out using a strain-controlled rheometer (RFS III, Rheometric Scientific, Piscataway, NJ, USA) equipped with a parallel plate geometry (gap 2 mm, f = 50 mm within the temperature range 20–60 °C) and a Peltier system (±0.1 °C) for temperature control. The rheological measurements allow us to determine the complex modulus (G*), which is a quantitative measure of material stiffness or resistance to deformation [53,54]. Following the mathematical representation that expresses G* as a complex number, its modulus could be expressed as the following:(1)$$\left|{G*}_{\omega }\right|=\sqrt{{G^{\prime }}_{\left(\omega \right)}{}^{2}+{G^{\prime \prime }}_{\left(\omega \right)}{}^{2}}$$
where G′ and G″ are, respectively, the real and the imaginary parts of G*, both of which could be determined directly by rheological measurement [53,54]. The elastic contribution to G* is called the storage modulus (G′) because it represents the storage of energy and the elastic response of the material under oscillation (ω). Conversely, the viscous or inelastic contribution is called the loss or dissipation modulus (G″) since it represents energy loss during the same deformation [55]. Dynamic experiments were conducted after the determination of the visco-elastic region: in particular, temperature ramp tests were performed using a temperature range between 20.0 °C and 60.0 °C in heating (rate of 1 °C/min).

### 3.4. Powder X-ray Diffraction

Powder X-ray diffraction patterns of natural CB, modified samples B and D, and PO, were acquired with a Bruker D2-Phaser equipped with a Cu Kα radiation (λ = 1.5418 Å) and a Lynxeye detector, at 30 kV and 10 mA, with a step size of 0.01°(2θ) between 5 and 40°(2θ) and an acquisition time of 0.1 s/step angle. Samples were dissolved at a temperature of ca. 70 °C, transferred in the sample holder, slowly cooled down to room temperature, and stored at 4 °C for 24 h, twice. Finally, they were dissolved for a third time at ca. 70 °C, cooled down to room temperature, and stored at 4 °C for 72 h before taking measurements. This procedure was necessary to ensure identical starting conditions for all the samples and avoid a lack of reproducibility in the measurements (mostly due to the solid-state polymorphism evidenced by both natural CB and PO [32,51]). The obtained patterns were analyzed using the Bruker Diffrac.Eva software, version V3.1.

### 3.5. Nuclear Magnetic Resonance (NMR)

All of the NMR experiments for molecular characterization were performed using a Bruker Avance 500 MHz spectrometer working at a field strength of 11.74 T (500 MHz ^1^H Larmor frequency), equipped with a 5 mm multinuclear probe TBO (triple-resonance broadband observe) and a standard variable-temperature control unit BVT-3000 (Bruker, Fällanden, Switzerland) [56,57,58]. Generally, 40.0 mg of sample were solubilized in deuterated chloroform (CDCl_3_-99.8 atom % D, contains 0.03 % (*v*/*v*) TMS; Sigma Aldrich, Milan, Italy), and transferred in a 5 mm NMR tube; trimethylsilane (TMS) was used for NMR spectra calibration as an internal reference standard for chemical shifts [56,57]. All of the isotropic spectra were recorded at room temperature. Spectral assignment of lipids was based on the one-dimensional (1D) ^1^H, ^13^C, ^13^C-{^1^H} NMR spectra, bi-dimensional (2D) homo and heteronuclear correlation NMR experiments (^1^H COSY, ^1^H-^13^C HMQC), compared with data from the literature. They allowed for a qualitative and quantitative characterization of the percentage of TAGs, and these data were also confirmed and integrated by GC/MS.

The ^1^H NMR experiments were acquired using a spectral width (SW) of 12.00 ppm, a relaxation delay (D1) of 6.0 s, and collecting 124 FIDs points and 128 scans. For 1D ^13^C-{^1^H} NMR spectra, recorded using proton broad-band decoupling (Bruker pulse sequence *zgdc*), 32K FIDs were collected using a SW of 250.00 ppm and a D1 of 10.0 s. Then, 1D NMR spectra were Fourier-transformed and manually phased, baseline-corrected, and aligned using the TMS signal as a reference. The ^13^C-{^1^H} NMR spectra were filtered with a small exponential multiplication (LB = 0.20 Hz) before Fourier transformation. The ^1^H COSY experiments were performed using a SW of 12.00 ppm in both dimensions, 2K data points, 40 scans, and 256 increments; the ^1^H-^13^C HMQC spectra were recorded using SWs of 14.00 ppm (^1^H) and 250.00 ppm (^13^C), 2K datapoints, 512 scans, and 40 experiments. A *sine* and a *qsine* filter were applied in both dimensions, F1 and F2, for the COSY and HMQC experiments, respectively, before being Fourier-transformed. The processing procedures on all spectra were carried out using TopSpin 3.6 software (Bruker Bio-Spin, Rheinstetten, Germany) (TopSpin, 2022) [59].

## 4. Conclusions

In this paper, several physical–chemical techniques were used to investigate the macroscopic and molecular properties of commercial CB and of three other modified samples obtained from it. The first important result obtained comes from rheological, DSC, and X-ray experiments: all these macroscopic investigations confirmed that the modified samples have a much more complicated polymorphic system than the native CB. This polymorphism and the high transition temperatures (up to about 53–54 °C) are difficult to explain by the existence in the modified samples of different crystalline forms due to different packing of the unique mixture of TAGs (i.e., SUS-type) existing in the natural CB. In order to rationalize this evidence, we hypothesized that the treatment to which the native fat was subjected had the effect of modifying the positional distribution of the acyl chains on the TAGs, which means that some of the fatty acids in the mixture of TAGs, mainly the SUS-types (i.e., POP, POS, SOS), modify their positional distribution with the formation of SSU-type (i.e., PPO, PSO, SPO, SSO). The evident differences in the DSC profiles, in the diffractometric patterns, and in the rheological measurements between the three modified samples (B–D), obtained under different experimental conditions, can be attributed to the different percentages of formation of the SSU-type TAGs.

Finally, the molecular study using ^13^C qNMR confirmed the hypothesis of the positional rearrangement of acyl chains that occurred during the modification process, and at the same time allows the quantification of the conversion degree for each sample.

Indeed, the visible differences in terms of chemical shifts and intensity of some peaks present in the ^13^C NMR spectra of the four samples, led us to conclude that the native CB mainly consists of SUS-type TAGs, while the modified fats consist of various mixtures of SUS-type and SSU-type TAGs. The relative conversion degree was obtained by integrating the pertinent peaks. It should be noted that in both ^1^H and ^13^C spectra of the modified butters there is no experimental evidence regarding the formation of other compounds or the formation of trans fatty acids. The key to these results is that natural and modified butters have the same composition in terms of percentage of single chains of FAs, but different positional distributions. This is fundamental to the food industry because this process manipulates only the distribution of fatty acids but not their nature, so the initial percentage of each FAs remains the same. Moreover, ^13^C qNMR has proved to be a valid and rapid tool for the regioisomeric investigation of fats and this opens up the possibility of using NMR spectroscopy as an alternative to the chromatographic techniques usually used for the characterization of similar matrices.

## Figures and Tables

**Figure 1 ijms-24-02090-f001:**
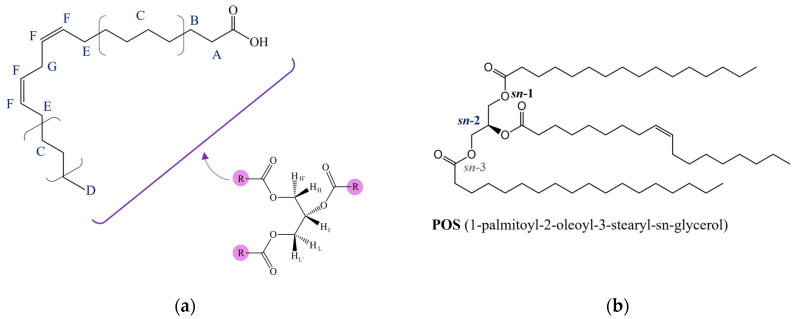
(**a**) General structure and proton nuclei labelling of molecular fragments in triacylglicerols (TAGs); (**b**) molecular structure of POS (1-palmitoyl-2-oleoyl-3-stearyl-sn-glycerol), the common SUS-type TAG recognized in CB.

**Figure 2 ijms-24-02090-f002:**
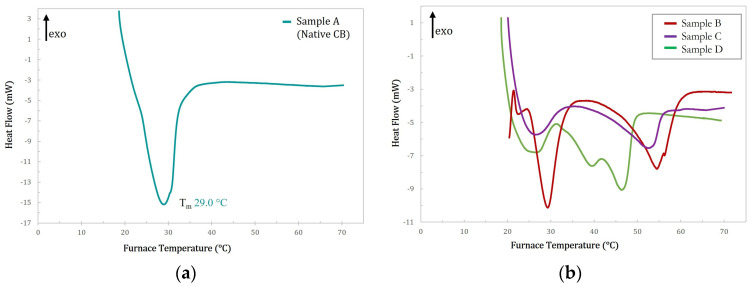
Melting profile (endo-down) of (**a**) native CB (sample A, blue curve) and (**b**) modified samples B (red curve), C (purple curve), and D (green curve).

**Figure 3 ijms-24-02090-f003:**
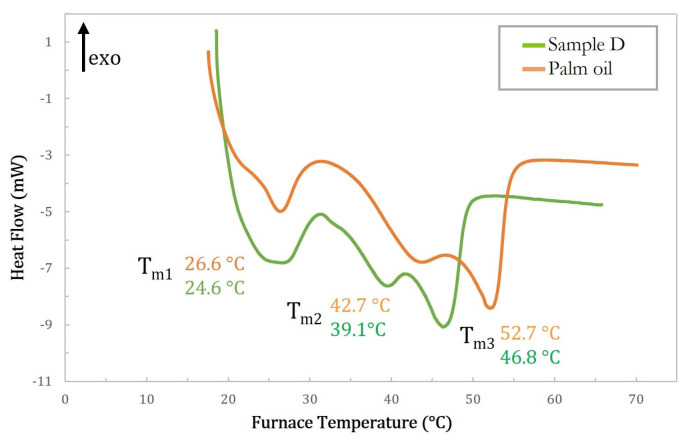
Melting profile of sample D (green curve) and PO (orange curve).

**Figure 4 ijms-24-02090-f004:**
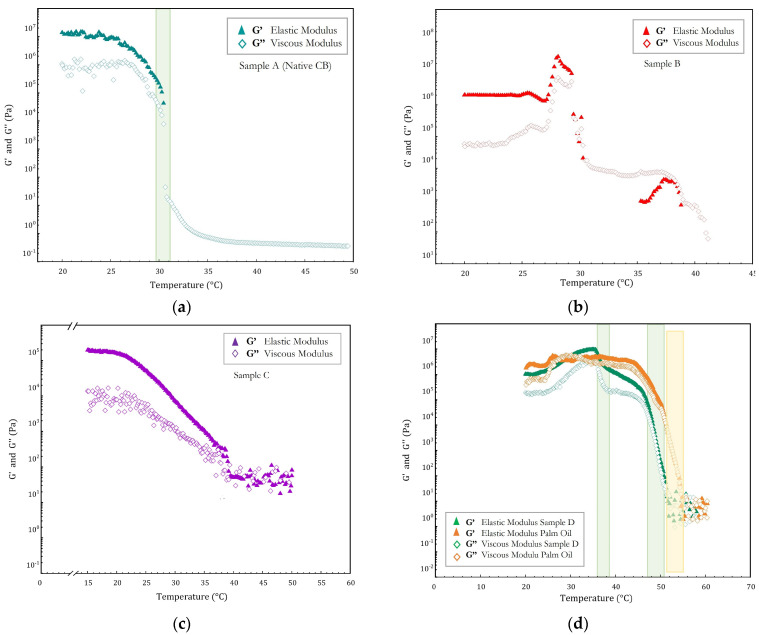
Time cure test (**a**) in the range 20–50 °C of native CB (sample A), of modified (**b**) sample B, and (**c**) sample C in the temperature range 15–50 °C. (**d**) Comparison of modified samples D (in green) with that of palm oil (PO, in orange) in the range 20–60 °C. Vertical lines define the temperature transition region.

**Figure 5 ijms-24-02090-f005:**
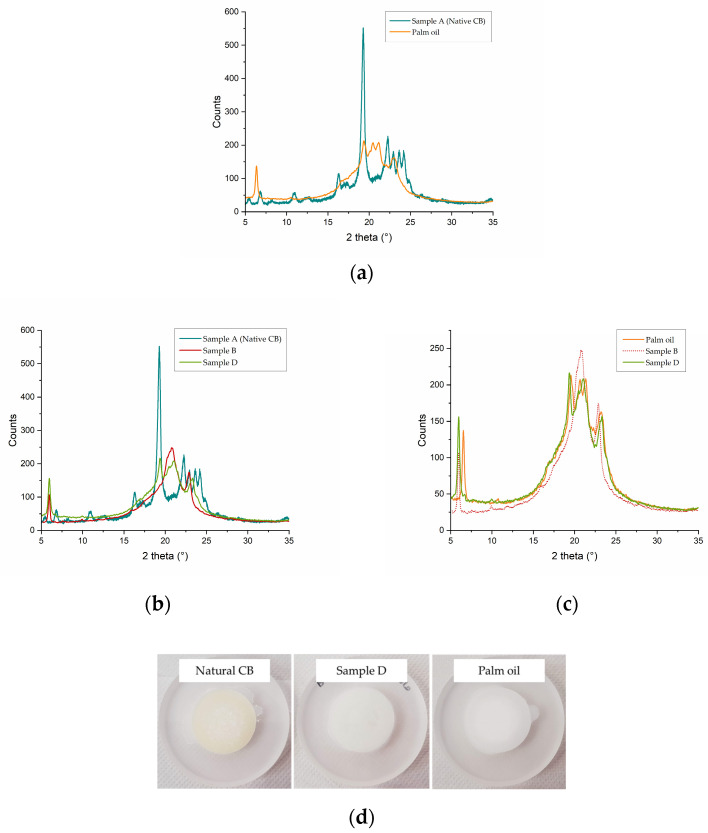
Comparison of PXRD patterns for (**a**) native CB (in blue), and PO (in orange); (**b**) native CB (in blue), sample B (in red), and sample D (in green); (**c**) PO (in orange), sample B (in red, dotted line) and sample D (in green). Figures show that the more heavily modified the fat, with SUS-type TAGs converted into SSU-type, the closer the X-ray profile is to that of PO. (**d**) A view of the native CB, modified sample D, and PO after the three cycles of heating/cooling and storage at 4 °C for 72 h before the PXRD measurements.

**Figure 6 ijms-24-02090-f006:**
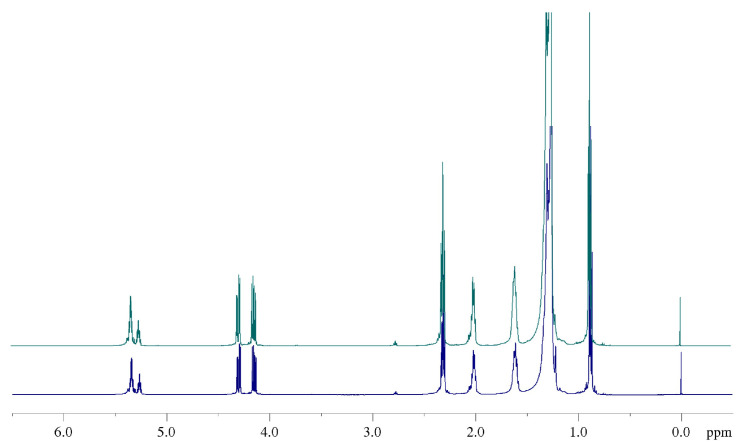
Comparison between ^1^H NMR spectra (500 MHz, 298 K, in CDCl_3_) of natural CB (sample A, blue line) and one of the modified fats (sample B, green line) to highlight the high similarity of the protonic profile.

**Figure 7 ijms-24-02090-f007:**
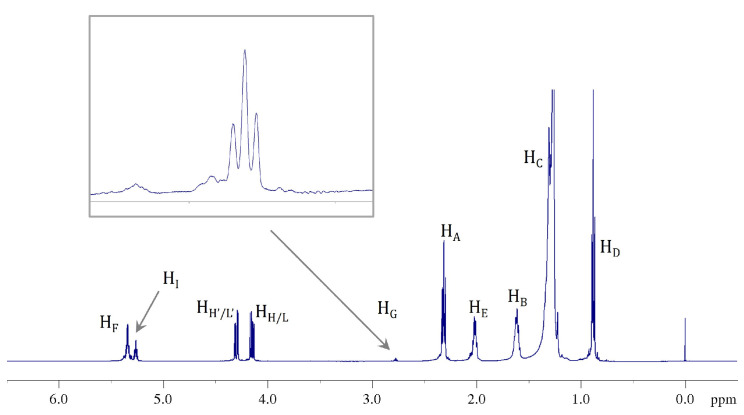
^1^H NMR spectrum (500 MHz, 298 K, in CDCl_3_) of CB (sample A) in which the signal attributions to the different functional groups of FAs in TAGs are shown.

**Figure 8 ijms-24-02090-f008:**
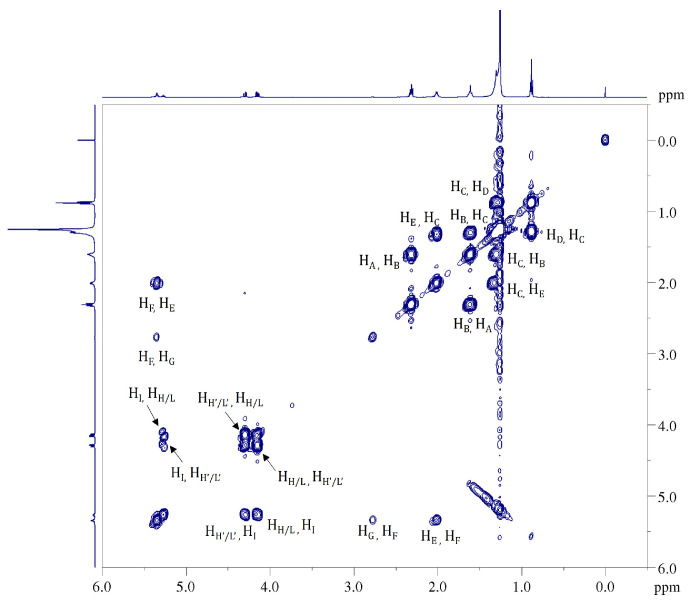
^1^H-^1^H COSY NMR spectrum (500 MHz, 298 K, in CDCl_3_) of natural CB (sample A). All cross peaks in the 2D map corresponding to the homuncular correlations among the protons in triacylglycerols (TAGs) have been highlighted.

**Figure 9 ijms-24-02090-f009:**
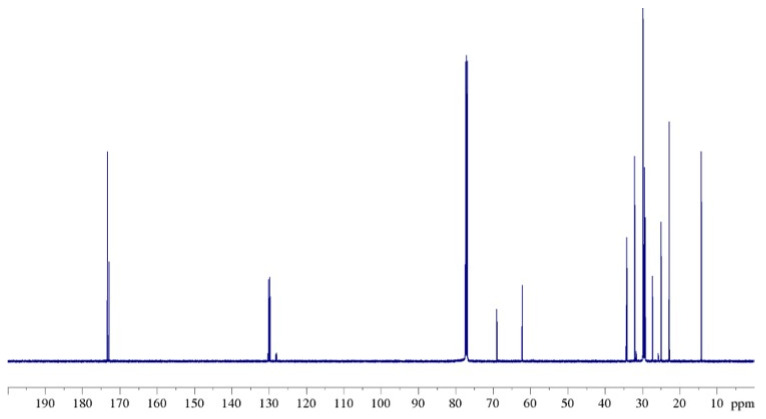
^13^C NMR spectrum (125 MHz, 298 K, in CDCl_3_) of natural CB (sample A).

**Table 1 ijms-24-02090-t001:** Temperature of melting ((T*_m_*, °C) and melting enthalpies (ΔH*_m_*, J/g) of native CB (sample A), modified samples B, C and D, and palm oil (PO) determined via DSC (Differential Scanning Calorimetry). Instrumental errors have been reported for both temperature and enthalpy values.

	$Tm_{\;}({}^{\circ}C,\;\pm 0.1)$	$\Delta Hm_{\;}(J/g,\;\pm 0.1)$
A-Native CB	29.0	53.5
B	28.6	20.8
	53.3	29.2
C	26.1	7.7
	52.6	11.0
D	24.6	10.7
	39.1	2.9
	46.8	9.6
PO	26.6	5.6
	42.7	5.1
	52.5	12.5

**Table 2 ijms-24-02090-t002:** ^1^H NMR chemical shifts (**δ_H_**, ppm) and ^1^H/^1^H correlations of fatty acid protons in triacylglycerols (TAGs) in CDCl_3_ for natural CB. The multiplicity of the signal was indicated where possible and the corresponding experimental *J*-couplings (Hz) were extracted from the spectrum. The letters shown below, which stand for protons belonging to different functional groups of FAs in TAGs, refer to the labelling reported for the first time in Figure 1a.

Position	δ_H_, Multiplicity ^1^ (*J* in Hz)	COSY
A	2.20–2.43, m	B
B	1.50–1.73, m	A, C
C	1.15–1.45, m	B, D, E
D	0.88, m	C
E	1.93–2.15, m	C, F
F	5.28–5.46, m (11.00)	E, G
G	2.73–2.81, (t, 6.5)	F
H, L (Gly ^1^)	4.14, dd (11.88, 5.93)	H′, L′, I
I (Gly ^1^)	5.26, m	H, H′, L, L′
H′, L′ (Gly ^1^)	4.29, dd (11.88, 4.31)	H, L, I

^1^ Abbreviations: d—doublet; dd—doublet of doublet; t—triplet; m—multiplet; Gly—Glycerol.

**Table 3 ijms-24-02090-t003:** ^13^C NMR (125 MHz, 298 K, in CDCl_3_) assignment for natural (sample A) and modified (sample B) CB. The values reported in the table show clearly how the chemical shifts (**δ**^13^C, ppm) for the FAs in CB depend often significantly on the position of the acyl chain on the glycerol scaffold (indicated as *sn*-1,2,3 as specified in the Introduction). The numbering for the carbon nuclei is explained in Figure 10.

Triglycerides		δ ^13^C ppm							
		Palmitic Acid		Stearic Acid		Oleic Acid		Linoleic Acid	
		A	B	A	B	A	B	A	B
C-1	*sn*-1,3	173.28	173.27	173.28	173.27	-	173.23	-	173.23
	*sn*-2	-	172.86	-	172.86	172.84	172.82	172.84	172.82
C-2	*sn*-1,3	34.08	34.08	34.08	34.08	-	34.05	-	34.05
	*sn*-2	-	34.24	-	34.24	34.22	34.22	34.22	34.22
C-3	*sn*-1,3	24.89	24.89	24.89	24.89	-	24.88	-	24.88
	*sn*-2	-	24.94	-	24.94	24.91	24.91	24.87	24.87
C-4	*sn*-1,3	29.16	29.16	29.16	29.16	-	28.12	-	
	sn-2	-	29.12	-	29.12	28.08	28.08	-	
C-5	*sn*-1,3	29.30		29.36	29.36	-	29.22	-	
	*sn*-2	-	29.33	-	29.36 (b ^1^)	29.23	29.23	-	
C-6	*sn*-1,3	29.51	29.51	29.51	29.51	-	29.14	-	29.14
	*sn*-2	-	29.53	-	29.53	29.15	29.15	29.15	29.15
C-7	*sn*-1,3	29.65	29.65	29.65	29.65	-	29.75 (b ^1^)	-	29.64
	*sn*-2	-	29.67	-	29.67	29.75	29.75	-	
C-8	*sn*-1,3	29.68–29.73				-	27.20	-	27.22
	*sn*-2					27.20		27.22	27.22
C-9	*sn*-1,3	29.68–29.73				-	128.72	-	129.98
	*sn*-2					128.72	128.72	-	
C-10	*sn*-1,3	29.68–29.73				-	130.02	-	128.08
	*sn*-2					130.03	130.03	128.11	128.11
C-11	*sn*-1,3	29.68–29.73				-	27.25		22.66
	*sn*-2					27.25	27.25	-	
C-12	*sn*-1,3	29.68–29.73				-	29.80	-	127.93
	*sn*-2					29.80	29.80	127.91	127.91
C-13	*sn*-1,3	29.40	29.40	29.68–29.73		-	29.35	-	130.21
	*sn*-2	29.68–29.73				29.35	29.35	130.23	130.23
C-14	*sn*-1,3	31.96	31.96	29.68–29.73		-	29.56 (b ^1^)	-	27.23
	*sn*-2	-	31.96	-	29.68–29.73	29.56		27.23	
C-15	*sn*-1,3	22.72	22.72	29.40	29.40	-	29.35	-	29.37
	*sn*-2	-	22.72	-	29.40	29.35	29.35	29.37	29.37
C-16	*sn*-1,3	14.13	14.13	31.96	31.96	-	31.94	-	31.55
	*sn*-2	-	14.13	-	31.96	31.94		31.55	
C-17	*sn*-1,3			22.71	22.72	-	22.71	-	22.60
	*sn*-2			-	22.72	22.60	22.60	-	
C-18	*sn*-1,3			14.13	14.13	-	14.12	-	14.08
	*sn*-2			-	14.12	14.08	14.08	-	-
-CHO (Gly ^1^)	*sn*-2	-	68.91	-	68.91	68.92			
-CH_2_O (Gly ^1^)	*sn*-1,3	62.11				-	62.12	-	62.12

^1^ Abbreviation: b-broad signal; Gly-Glycerol.

**Table 4 ijms-24-02090-t004:** ^13^C qNMR regiospecific analysis for samples A (natural CB), B, C, and D. Because of only a relative and not an absolute quantitation via NMR was conducted, the composition in terms of unsaturated or saturated FAs in a stereospecific position is reported as a percentage (%). ^13^C NMR results are the mean of three replicates. Each integration was carried out considering the carbonyl (C-1) and α-carbons (C-2) signals.

	Composition (%; ±σ)			
	Unsaturated		Saturated	
	*sn*-1,3	*sn*-2	*sn*-1,3	*sn*-2
Sample A		32.7 ± 0.3	67.3 ± 0.2	-
Sample B	15.2 ± 0.4	17.7 ± 0.4	30.2 ± 0.2	36.7 ± 0.2
Sample C	13.3 ± 0.6	20.1 ± 0.3	40.5 ± 0.3	25.7 ± 0.5
Sample D	21.4 ± 0.3	10.8 ± 0.5	20.2 ± 0.6	47.1 ± 0.4

## Data Availability

All data generated or analyzed during this study are included in this published article.

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
