# Peer review of "Triacylglycerol Composition and Chemical-Physical Properties of Cocoa Butter and Its Derivatives: NMR, DSC, X-ray, Rheological Investigation"

_ijms, 2023, doi:10.3390/ijms24032090_

Round 1
Reviewer 1 Report
Manuscript title: Triacylglycerol composition and chemical-physical properties of Cocoa Butter and its derivatives: NMR, DSC, X-Ray, rheological investigation
This study has certain significance in food chemistry of cocoa butter ……………... However, revisions are necessary for the current version of the manuscript. The following questions to be addressed/considered may be helpful to improve the manuscript.
Major comments
· Insufficient Abstract: In the abstract, the main aim and background of the manuscript are missing, the current version it only highlights the result. In addition, it would be even better to have a sentence as a future perspective.
· The unit/abbreviation is not mentioned before, consider defining the abbreviation when mentioned for the first time…. Please check throughout the manuscript to define the abbreviations.
· Lake of scientific literature to support the statements and findings throughout the manuscript…... I have made some suggestions for that and more need it….
· More information is needed for ALL TABLE captions and define the abbreviation and units that are used. And adjust the significant figures for the table and manuscript.
· Grammar and punctuation issuers need to be addressed. I have selected/mentioned some as examples.
· I am not sure whether the ‘’…..’’ term is well discussed in the abstract and manuscript. Please consider discussing it or rephrasing it.
· I have a major concern about the results and discussion section. The authors describe the results and compare the results with previous studies, however, insight mechanisms are still insufficient.
· The language is generally clear, with some exceptions where the authors are a bit too innovative with the terminology, and made very long sentences …..
Detailed comments:
Abstract
Line 12-17: A complicated sentence, please revise and check the grammar
Introduction:
Line 36-40: a long sentence.
Line 41-44: A reference is needed here.
Line 50-64: A reference is needed here.
Line 66-74: A complicated sentence, please revise and check the grammar
In MM section
Literature references are missing for all sub-section. It would be better to cite the references that the procedure adopted.
Additional info is needed for the table caption, most importantly significant figures.
In MM section, what is the quality control (QC) data? There is no mention of the QC.
What is the accuracy of the instruments, recovery, LOD, and LOQ ……. These parameters are needed to report the efficiency of any analytical system.
In general, how many times you’ve recorded the data,? duplicate? Triplicate?..... what you mentioned in the text is not clear, please elaborate more on this
R&D section
These sections are repeating information already presented and explain things in an unnecessarily complicated way. The quality of the manuscript would benefit from the whole section being condensed, Line 201-225, Line271-292, Line 297-321, Line 359-424.
Line 46-47: A reference is needed here, for example, you can use: https://onlinelibrary.wiley.com/doi/book/10.1002/9781118788745
Line 371-385: A reference is needed here, for example, you can use: https://doi.org/10.1007/s12161-021-02203-0
Figure 11. There is a typo error, (d) repeated5t two times.
Author Response
Rende, January 10, 2023
Dear reviewer,
thank you very much for the effort that you spent on our manuscript. We appreciate the very constructive and helpful comments and suggestions to improve our work. In response to your input, we modified the text's recommended modifications.
We hope that our revised manuscript addresses all concerns satisfactorily.
All changes are presented in detail below. The issues raised by the reviewer are set in italics and our answers are in plain font. All our changes are included in the revised manuscript in red color.
ANSWERS TO REFEREE 1
Open Review
(x) I would not like to sign my review report
( ) I would like to sign my review report
English language and style
( ) Extensive editing of English language and style required
(x) Moderate English changes required
( ) English language and style are fine/minor spell check required
( ) I don't feel qualified to judge about the English language and style
|
Yes |
Can be improved |
Must be improved |
Not applicable |
|
|
Does the introduction provide sufficient background and include all relevant references? |
( ) |
( ) |
(x) |
( ) |
|
Are all the cited references relevant to the research? |
( ) |
( ) |
(x) |
( ) |
|
Is the research design appropriate? |
( ) |
(x) |
( ) |
( ) |
|
Are the methods adequately described? |
( ) |
( ) |
(x) |
( ) |
|
Are the results clearly presented? |
(x) |
( ) |
( ) |
( ) |
|
Are the conclusions supported by the results? |
(x) |
( ) |
( ) |
( ) |
Comments and Suggestions for Authors
Manuscript title: Triacylglyol composition and chemical-physical properties of Cocoa Butter and its derivatives: NMR, DSC, X-Ray, rheological investigation
This study has certain significance in food chemistry of cocoa butter ……………... However, revisions are necessary for the current version of the manuscript. The following questions to be addressed/considered may be helpful to improve the manuscript.
Major comments
Reviewer: Insufficient Abstract: In the abstract, the main aim and background of the manuscript are missing, the currentversion it only highlights the result. In addition, it would be even better to have a sentence as a future perspective.
Authors: We are grateful to the reviewer and following its suggestion we reworked the entire abstract highlighting the purpose of the work, its importance for industrial research in the context of future applications.
Then, the text of the abstract at page 1, from line 12 to line 29 (of the original manuscript): “Nowadays, having vegetable fats and oils, among the main ingredients in pastry, with a well-defined, specific lipidic composition and appropriate mechanical properties (ease of transport, processing, and storage) is one of the most urgent requests of industrial food research, more and more committed to giving healthy products to consumers. The chemical-physical behavior of these matrices does not depend only on their chemical composition (i.e., the type and the percentage of Fatty Acids (FAs) present), neither is this capable to ensure the uptake of specific FAs in humans. As we demonstrate in this work, these properties are also largely determined by their regiosomerism within the TriAcylGlycerols (TAGs) moieties (sn-1,2,3 positions). The focus of this work is the study of some vegetable fats obtained directly from a sample of natural Cocoa Butter (CB), for the first time, through a process that manipulates the distribution of FAs but not their nature, so that the initial percentage of each FAs in the mixture remains the same. In order to understand which factors would account for their physical and chemical characteristics and check whether or not these obtained new matrices could be considered as valid alternative to other vegetable fats (i.e. palm oil) an experimental investigation has been carried out, both at macroscopic and molecular level, including: i) DSC analyses to examine thermal features; ii) rheological testing to explore mechanical properties; iii) powder X-Ray diffraction to evaluate the solid-state phases of the ob-tained fats; iv) 1H and 13C NMR (1D and 2D) spectroscopy to analyze fatty acid composition in-cluding regioisomeric distribution on the glycerol backbone.”
Have been replaced by:
“Nowadays food industry is increasingly involved in researching vegetable fats and oils with appropriate mechanical properties (ease of transport, processing, and storage) and a specific lipidic composition mainly to ensure healthy products for consumers. However, the chemical-physical behavior of these matrices depends on their composition in terms of single Fatty Acids (FA) but, as we demonstrate in this work, these properties, as well as the absorption, digestion and uptake in humans of specific FAs, are also largely determined by their regiosomerism within the TriAcyl-Glycerols (TAG) moieties (sn-1,2,3 positions). The goal of this work is the study of some vegetable fats obtained directly from a sample of natural Cocoa Butter (CB), for the first time, through a process that manipulates the distribution of FAs but not their nature: even if the initial percentage of each FAs in the mixture remains the same, CB derivatives seems to show improved chemi-cal-physical features. In order to understand which factors would account for their physical and chemical characteristics and check whether or not these obtained new matrices could be considered as valid alternative to other vegetable fats (i.e. palm oil (PO)) an experimental investigation has been carried out, both at macroscopic and molecular level, including: i) Differential Scanning Calorimetry (DSC) analyses to examine thermal features; ii) rheological testing to explore mechanical properties; iii) powder X-Ray diffraction (PXRD) to evaluate the solid-state phases of the obtained fats; iv) 1H and 13C Nuclear Magnetic Resonance (NMR, 1D and 2D) spectroscopy to analyze rapidly fatty acid composition including regioisomeric distribution on the glycerol backbone. These last results open up the possibility of using NMR spectroscopy as an alternative to the chromato-graphic techniques routinely employed for the investigation of similar matrices.”
Moreover, to better clarify the aim of the work, at page 3 of the Introduction section, from line 98 to line 113 (original manuscript), the sentence “In this work, an experimental investigation has been carried out on the chemical-physical properties of both natural CB and some modified butters obtained from it thought a process acting on the FAs position, but not on their nature, within the naturally present TAGs. The TAG composition, crystallization, thermal behavior, and rheological properties have been investigated on all the materials to better understand at molecular and macroscopic level their characteristics and properties. The experimental investigation carried out include: i) DSC tests to examine thermal features; ii) rheological testing to explore mechanical properties; iii) powder X-Ray diffraction to qualitatively compare the obtained matrices at the solid-state level; iv) 1H and 13C NMR (1D and 2D) spectroscopy for a molecular characterization. The accordance between experimental results from these different techniques confirms how a specific chemical and physical behavior is strongly determined by the stereochemistry of TAGs. Particularly NMR allows a qualitative and quantitative (in percentage of TAGs) characterization of these complex mixtures and provides useful information on the positional distribution of FAs.
Have been replaces by:
“Thus, in this context, the present study was designed to experimentally investigate the changes in the chemical-physical properties of some modified fats derived from a natural CB sample through an unconventional process. The aim is to demonstrate that this modification process has led to a manipulation of the positional distribution of the FAs but has left their nature unaltered, i.e. following the modification process, positional isomers have been obtained in which part of the FAs in the mixture of SUS-type TAGs (mainly POP, POS and SOS) were converted into SSU-type (Saturated-Saturated-Unsaturated- type) mainly PPO (1,2-palmitoyl-3-oleoyl-sn-glycerol), PSO (1-palmitoyl-2- stearyl -3-Oleoyl-sn-glycerol), SPO (1-stearyl-2- palmitoyl -3-oleoyl-sn-glycerol) and SSO (1,2-stearyl-3-oleoyl-sn-glycerol). To this end, a comparative study among the native CB sample, three samples of fats derived from it and a sample of PO was carried out. The TAG composition, crystallization, thermal behavior and rheological properties have been studied on all these materials in order to understand their characteristics, properties and differences at the molecular and macroscopic level. In particular, the experimental investigation carried out included: i) Differential Scanning Calorimetry (DSC) tests to examine thermal features (mainly melting points); ii) rheological testing to explore mechanical properties; iii) powder X-Ray diffraction (PXRD) to qualitatively compare the obtained matrices at the solid-state level; iv) 1H and 13C Nuclear Magnetic Resonance (NMR, 1D, and 2D) spectroscopy for molecular characterization. It should be emphasized that the NMR technique was particularly useful in the qualitative and quantitative (in the percentage of TAGs) characterization of these complex mixtures and, basically, it allowed to experimentally confirm the positional rearrangement of the acyl chains that occurred during the modification process.The accordance between experimental results from these different techniques confirms how a specific chemical and physical behavior is strongly determined by the stereochemistry of TAGs.”
Reviewer: The unit/abbreviation is not mentioned before, consider defining the abbreviation when mentioned for the first time…. Please check throughout the manuscript to define the abbreviations.
Authors: We are grateful to the reviewer for noting this as so that we can make the necessary corrections. All the new abbreviation, defined when the related expressions are reported for the first time, are now reported in red in the new manuscript.
Reviewer: Lake of scientific literature to support the statements and findings throughout the manuscript…... I have made some suggestions for that and more need it….
Authors: Following the suggestions of the reviewer, we have added new references, including those indicated by the referee and reported in the minor comments below. In particular we added:
- In the “Introduction” section and then in the “References” section:
[3] Pande, G.; Akoh, C. C.; Lai, O. M. Food Uses of Palm Oil and Its Components. In Palm Oil: Production, Processing, Characterization, and Uses, 1st ed.; Lai, O.-M., Tan, C.-P., Akoh, C.C., Eds.; AOCS Press, Urbana, Illinois, USA, 2012; pp. 561-586.
[4] Mba, O. I.; Dumont, M. J.; Ngadi, M. Palm oil: Processing, characterization, and utilization in the food industry - A review. Food Biosci., 2015, 10, 26-41. https://doi.org/10.1016/j.fbio.2015.01.003
[5] Wijewardene, U.; Premlal Ranjith, H. M. Food safety and quality issues of dairy fats. In Fats in Food Technology, 2nd ed.; Kanes K. Rajah Ed.; John Wiley & Sons, 2014; pp. 289-325. https://doi.org/10.1002/9781118788745.ch8
[6] Abd-Aziz, S.; Gozan, M.; Ibrahim, M.F.; Phang, L.-Y. Demand and Sustainability of Palm Oil Plantation. In Biorefinery of Oil Producing Plants for Value-Added Products, 1st ed.; Abd-Aziz, S., Gozan, M., Ibrahim, M.F., Phang, L.-Y. Eds.; Wiley Online Library, 2022. https://doi.org/10.1002/9783527830756.ch2
[7] Fitzherbert, E. B.; Struebig, M. J.; Morel, A.; Danielsen, F.; Brühl, C. A.; Donald, P. F.; Phalan, B. How will oil palm expansion affect biodiversity? Trends Ecol. Evol. 2008, 10, 538-545. https://doi.org/10.1016/j.tree.2008.06.012
[11] Fadda, A.; Sanna, D.; Sakar, E. H.; Gharby, S.; Mulas, M.; Medda, S.; Durazzo, A. Innovative and sustainable technologies to enhance the oxidative stability of vegetable oils. Sustainability, 2022, 14(2), 849. https://doi.org/10.3390/su14020849
[12] List, G.R.; Wang, T.; Shukla, V.K.S.; (2023). Storage, Handling, and Transport of Oils and Fats. In Bailey's Industrial Oil and Fat Products; Shahidi F. Ed.; Wiley Online Library, 2023. https://doi.org/10.1002/047167849X.bio049.pub2
[13] Nunes, A.; Marto, J.; Gonçalves, L.; Martins, A. M.; Fraga, C.; Ribeiro, H. M. Potential therapeutic of olive oil industry by‐products in skin health: a review. Int. J. Food Sci. Technol., 2022, 57(1), 173-187. https://doi.org/10.3390/su14020849
[14] Szabo, Z.; Marosvölgyi, T.; Szabo, E.; Koczka, V.; Verzar, Z.; Figler, M.; Decsi, T. Effects of Repeated Heating on Fatty Acid Composition of Plant-Based Cooking Oils. Foods, 2022, 11(2), 192. https://doi.org/10.3390/foods11020192
[15] Nieto, G.; Lorenzo, J. M., Chapter 4: Plant source: Vegetable oils. In Food Lipids, 1st ed.; José, M.L., Sichetti Munekata, P.E., Pateiro, M., Barba, F.J., Domínguez, R., Eds.; Academic Press, 2022, pp. 69-85. https://doi.org/10.1016/B978-0-12-823371-9.00011-3
[17] Norazlina, M.R.; Jahurul, M.H.A.; Hasmadi, M.; Mansoor, A.H.; Norliza, J.; Patricia, M.; Fan, H.Y. Trends in blending vegetable fats and oils for cocoa butter alternative application: A review. Trends Food Sci Technol., 2021, 116, 102-114. https://doi.org/10.1016/j.tifs.2021.07.016
- In the section “2. Matherial and Method” and then in the “References” section:
[30] Bresson, S.; Lecuelle, A.; Bougrioua, F.; El Hadri, M.; Baeten, V.; Courty, M.; Faivre, V. Comparative structural and vibrational investigations between cocoa butter (CB) and cocoa butter equivalent (CBE) by ESI/MALDI-HRMS, XRD, DSC, MIR and Raman spectroscopy. Food Chem., 2021, 363, 130319. https://doi.org/10.1016/j.foodchem.2021.130319
[31] Hadri, M.E.; Bresson, S.; Lecuelle, A.; Bougrioua, F.; Baeten, V.; Nguyen, V.H.; Courty, M. Structural and Vibrational Investigations of Mixtures of Cocoa Butter (CB), Cocoa Butter Equivalent (CBE) and Anhydrous Milk Fat (AMF) to Understand Fat Bloom Process. Appl. Sci., 2022, 12(13), 6594. https://doi.org/10.3390/app12136594
- In the section “Rheological Measurement “and then in the “References” section:
[33] Malkin, A.Y.; IIsayev, A. Rheometry Experimental Methods. In Rheology Concepts, Methods, and Applications, 2nd ed.; Malkin, A.Y.; IIsayev, A., Eds.; Elsevier, 2012; pp. 255-364. https://doi.org/10.1016/B978-1-895198-49-2.50010-4
[34] Joyner (Melito), H.S.; Daubert, C.R. Rheological Principles for Food Analysis. In Food Analysis, 1st ed.; Nielsen, S.S., Ed.; Food Science Text Series. Springer, Cham, 2017; pp. 511-527. https://doi.org/10.1007/978-3-319-45776-5_29
- In the section “Nuclar Magnetic Resonance (NMR) “and then in the “References” section:
[36] [Reuhs, B.L.; Simsek, S. Nuclear Magnetic Resonance. In Food Analysis, 1st ed.; Nielsen, S.S., Ed.; Food Science Text Series. Springer, Cham, 2017; pp. 151-163. https://doi.org/10.1007/978-3-319-45776-5_10
[37] Truzzi, E.; Marchetti, L.; Benvenuti, S.; Ferroni, A.; Rossi, M.C.; Bertelli, D. Novel strategy for the recognition of adulterant vegetable oils in essential oils commonly used in food industries by applying 13C NMR spectroscopy. J. Agric. Food Chem., 2021, 69(29), 8276-8286. https://doi.org/10.1021/acs.jafc.1c02279
[38] Eisenmann, P.; Ehlers, M.; Weinert, C.H.; Tzvetkova, P.; Silber, M.; Rist, M.J.; Luy, B.; Muhle-Goll, C. Untargeted NMR Spectroscopic Analysis of the Metabolic Variety of New Apple Cultivars. Metabolites 2016, 6(3):29. https://doi.org/10.3390/metabo6030029
- In the section “13C NMR of Natural and modified Cocoa Butter. A parallel study “and then in the “References” section:
[55] Hama, J.R.; Fitzsimmons-Thoss, V. Determination of Unsaturated Fatty Acids Composition in Walnut (Juglans regia L.) Oil Using NMR Spectroscopy. Food Anal. Methods, 2022, 15, 1226-1236. https://doi.org/10.1007/s12161-021-02203-0]
Reviewer: More information is needed for ALL TABLE captions and define the abbreviation and units that are used. And adjust the significant figures for the table and manuscript.
Authors: Thanks for careful comments. Following these tips, we corrected the significant figures in the text and also in Table 4 and we rewrote the table headers adding more details. Particularly:
- “Table 1. Temperature of melting and melting enthalpies of native CB and modified samples B, C and D, determined via DSC (Differential Scanning Calorimetry). Instrumental errors been reported for the Temperature (Tm)and Enthalpy (ΔHm)”
Has been replaced by:
“Table 1. Temperature of melting ((Tm, °C) and melting enthalpies (ΔHm, J/g) of native CB (sample A), modified samples B, C and D, and palm oil (PO) determined via DSC (Differential Scanning Calorimetry). Instrumental errors have been reported for both Temperature and Enthalpy values.”
- Table 2. 1H NMR chemical shifts and 1H/1H correlations of fatty acids protons in triacylglicerols (TAGs) in CDCl3 for natural cocoa butter
Has been replaced by:
“Table 2. 1H NMR chemical shifts (δH, ppm) and 1H/1H correlations of fatty acids protons in triacylglycerols (TAGs) in CDCl3 for natural CB. The multiplicity of the signal was indicated where possible and the corresponding experimental J-couplings (Hz) were extracted from the spectrum. The letters shown below, which stand for protons belonging to different functional groups of FAs in TAGs, refer to the labelling reported for the first time in Figure 1.a.”
- Table 3. 13C NMR assignment for natural (sample A) and modified (sample B) Cocoa Butter (500 MHz, 298 K, in CDCl3).
Has been replaced by:
“Table 3. 13C NMR (125 MHz, 298 K, in CDCl3) assignment for natural (sample A) and one of the modified (sample B) CB. The values reported in the tables shows clearly how the chemical shifts (δ 13C, ppm) for all the mainly FAs in CB depend often significantly by the position of the acyl chain on the glycerol scaffold (indicated as sn-1,2,3 as specified in the Introduction). The numbering for the carbon nuclei in referred to that explained in Figure 10. “
- Table 4. 13C qNMR regiospecific analysis for sample A, B, C and D. 13C NMR results are the mean of three replicates: for each integration was carried out considering the carbonyl (C-1) and α-carbons (C-2) signals.
Has been replaced by:
“Table 4. 13C qNMR regiospecific analysis for samples A (natural CB), B, C, and D. Because of only a relative and not an absolute quantitation via NMR has been conducted, the composition in term of Unsaturated or Saturated FAs in a stereospecific position in reported as a percentage (%). 13C NMR results are the mean of three replicates: for each integration was carried out considering the carbonyl (C-1) and α-carbons (C-2) signals”.
Reviewer: Grammar and punctuation issuers need to be addressed. I have selected/mentioned some as examples.
Authors: We have carefully checked all possible grammar and punctuation errors throughout the manuscript. All changes are shown in red and we hope this time we were more precise.
Reviewer: I am not sure whether the ‘’…..’’ term is well discussed in the abstract and manuscript. Please consider discussing it or or rephrasing it.
Authors: As we answered earlier, we have rephrased the abstract and now we hope it fits well.
Reviewer: I have a major concern about the results and discussion section. The authors describe the results and compare the results with previous studies; however, insight mechanisms are still insufficient.
Authors: We are grateful to the reviewe for the remark. As we specified in the text, we try to razionalize the experimental results and the chemical-Physical behaviour of natural CB and modied fats that emerges from all the techniques used for this study with the hypothesis of a molecular rearrangment of Tryacilglicerols. In particular this mechanism consists of a displacement of the Fatty Acids chains on the glycerol backbone to make the formations of positional isomers of the starting TAGs. This is a purely statistical rearrangement and the quantity of new TAGs generated by the process depend also by the reaction condition. It is therefore a classic interesterification reaction which leads our modified samples being mixtures of SUS-type (Saturated- Unsaturated- Saturated type, i.e., POP, POS, SOS) TAGs and also SSU-type TAGs (Saturated- Saturated- Unsturated i.e., PPO, PSO, SSO). However, as we spacified in the section “2.1 Starting Material and Modified Fats” our modification process was “carried out without enzymes or chemical additives conventionally employed [28-29]. In this case, natural CB was treated by appropriately regulating temperature and pressure under a controlled atmosphere. The three samples B, C, and D were obtained under different experimental conditions, but since the process will be covered by an industrial patent for obvious reasons of secrecy, it is not possible to reveal precise details on the three processes.” So other details about operating conditions cannot be revealed.
Reviewer: The language is generally clear, with some exceptions where the authors are a bit too innovative with the terminology, and made very long sentences.
Authors: We have reworked many long sentences and corrected the grammar as suggested by the review. All the corrections are reported in red in the new manuscript.
Detailed comments:
Reviewer: Abstract Line 12-17: A complicated sentence, please revise and check the grammar
Authors: As we discussed above under “Major comments”, the abstract section was completely rearranged to highlighting the purpose, the importance for industrial research and the future perspectives of this type of study.
Reviewer: Introduction: Line 36-40: a long sentence.
Author: We followed the reviewer's suggestion and the sentence at pag. 1, from line 36 to line 40 (original manuscript): “FAs are mainly hydrocarbon biomolecules, generally with an even number of carbon atoms, a carboxylic functional group at one end of the chain and a variable amount of π-bonds such that it is possible to talk about Saturated (SFAs), Mono-Unsaturated or Monoenoic (MUFAs) and Polyunsaturated (PUFAs) Fatty Acids [2].”
Has been replaced by:
“FAs are mainly hydrocarbon biomolecules, generally with an even number of carbon atom and a carboxylic functional group at one end of the chain. According to the number of π-bonds found in the acyl chain it is possible to talk about Saturated (SFAs), Mono-Unsaturated or Monoenoic (MUFAs), and Polyunsaturated (PUFAs) Fatty Acids [2].”
Reviewer: Line 41-44: A reference is needed here.
Author: Thank you for the suggestion. We introduce two new reference here:
[3] Pande, G.; Akoh, C. C.; Lai, O. M. Food Uses of Palm Oil and Its Components. In Palm Oil: Production, Processing, Characterization, and Uses, 1st ed.; Lai, O.-M., Tan, C.-P., Akoh, C.C., Eds.; AOCS Press, Urbana, Illinois, USA, 2012; pp. 561-586.
[4] Mba, O. I.; Dumont, M. J.; Ngadi, M. Palm oil: Processing, characterization, and utilization in the food industry - A review. Food Biosci., 2015, 10, 26-41. https://doi.org/10.1016/j.fbio.2015.01.003
Reviewer: Line 50-64: A reference is needed here.
Author: Thank you for the suggestion. References introduced in this section are reported below:
[11] Fadda, A.; Sanna, D.; Sakar, E. H.; Gharby, S.; Mulas, M.; Medda, S.; Durazzo, A. Innovative and sustainable technologies to enhance the oxidative stability of vegetable oils. Sustainability, 2022, 14(2), 849. https://doi.org/10.3390/su14020849
[12] List, G.R.; Wang, T.; Shukla, V.K.S.; (2023). Storage, Handling, and Transport of Oils and Fats. In Bailey's Industrial Oil and Fat Products; Shahidi F. Ed.; Wiley Online Library, 2023. https://doi.org/10.1002/047167849X.bio049.pub2
[13] Nunes, A.; Marto, J.; Gonçalves, L.; Martins, A. M.; Fraga, C.; Ribeiro, H. M. Potential therapeutic of olive oil industry by‐products in skin health: a review. Int. J. Food Sci. Technol., 2022, 57(1), 173-187. https://doi.org/10.3390/su14020849
[14] Szabo, Z.; Marosvölgyi, T.; Szabo, E.; Koczka, V.; Verzar, Z.; Figler, M.; Decsi, T. Effects of Repeated Heating on Fatty Acid Composition of Plant-Based Cooking Oils. Foods, 2022, 11(2), 192. https://doi.org/10.3390/foods11020192
[15] Nieto, G.; Lorenzo, J. M., Chapter 4: Plant source: Vegetable oils. In Food Lipids, 1st ed.; José, M.L., Sichetti Munekata, P.E., Pateiro, M., Barba, F.J., Domínguez, R., Eds.; Academic Press, 2022, pp. 69-85. https://doi.org/10.1016/B978-0-12-823371-9.00011-3
[17] Norazlina, M.R.; Jahurul, M.H.A.; Hasmadi, M.; Mansoor, A.H.; Norliza, J.; Patricia, M.; Fan, H.Y. Trends in blending vegetable fats and oils for cocoa butter alternative application: A review. Trends Food Sci Technol., 2021, 116, 102-114. https://doi.org/10.1016/j.tifs.2021.07.016
Reviewer: Line 66-74: A complicated sentence, please revise and check the grammar
Author: We followed the reviewer's suggestion and the sentence at pag. 2, from line 66 to line 78 (original manuscript): “Then, understanding by an appropriate experimental characterization how the chemical composition of these fats can influence their chemical and physical behavior should be of great interest to the industrial research. In this field, CB is, for example, an interesting subject matter in food Science as well as an essential ingredient in baking industry, responsible for the texture, gloss, and typical melting behavior of chocolate [10,11]. This vegetable fat is solid at room temperature because it is largely composed (over 70%) of symmetrical triglycerides with an Unsaturated (U) fatty acid at the sn-2 position, in particular oleic acid (C18:1), exhibiting a single double carbon bond starting at chain position nine (ω-9). Another typical unsaturated fatty acid present in the sn-2 position of this fat is linoleic acid (C18:2) having two double bonds starting at carbon 6 (ω-6). Saturated (S) FAs (mostly palmitic (C16:0) and stearic (C18:0)) are, instead, in the sn-1 and sn-3 positions of glycerol backbone.”
Has been replaced by:
“That is why industrial research has a great interest in understanding how the chemical composition of these fats can influence their chemical and physical behavior. For example, CB is an interesting subject matter in food science and an essential ingredient in the baking industry: it is responsible for the texture, gloss, and typical melting behavior of chocolate [21,22]. It is solid at room temperature because is largely composed (over 70%) of symmetrical triglycerides with an Unsaturated fatty acid (generally indicated with the letter “U”) at the sn-2 position. These include principally Oleic acid (O, C18:1) and Linoleic acid (L, C18:2) in a much lower percentage. Saturated FAs (indicated with the letter “S” and mostly palmitic (P, C16:0) and stearic (S, C18:0)) are, instead, in the sn-1 and sn-3 positions of the glycerol backbone. This type of TAG is generally indicated as SUS (Saturated-Unsaturated-Saturated).”
Reviewer: In MM section. Literature references are missing for all sub-section. It would be better to cite the references that the procedure adopted.
Authors: The following references have been introduced in the various subsections of “Materials and Methods”
Subsections: Differential Scanning Calorimetry and Powder X-Ray Diffraction
[30] Bresson, S.; Lecuelle, A.; Bougrioua, F.; El Hadri, M.; Baeten, V.; Courty, M.; Faivre, V. Comparative structural and vibrational investigations between cocoa butter (CB) and cocoa butter equivalent (CBE) by ESI/MALDI-HRMS, XRD, DSC, MIR and Raman spectroscopy. Food Chem., 2021, 363, 130319. https://doi.org/10.1016/j.foodchem.2021.130319
[31] Hadri, M.E.; Bresson, S.; Lecuelle, A.; Bougrioua, F.; Baeten, V.; Nguyen, V.H.; Courty, M. Structural and Vibrational Investigations of Mixtures of Cocoa Butter (CB), Cocoa Butter Equivalent (CBE) and Anhydrous Milk Fat (AMF) to Understand Fat Bloom Process. Appl. Sci., 2022, 12(13), 6594. https://doi.org/10.3390/app12136594
Subsection: Rheological Measurement
[33] Malkin, A.Y.; IIsayev, A. Rheometry Experimental Methods. In Rheology Concepts, Methods, and Applications, 2nd ed.; Malkin, A.Y.; IIsayev, A., Eds.; Elsevier, 2012; pp. 255-364. https://doi.org/10.1016/B978-1-895198-49-2.50010-4
[34] Joyner (Melito), H.S.; Daubert, C.R. Rheological Principles for Food Analysis. In Food Analysis, 1st ed.; Nielsen, S.S., Ed.; Food Science Text Series. Springer, Cham, 2017; pp. 511-527. https://doi.org/10.1007/978-3-319-45776-5_29
Subsection: Nuclear Magnetic Resonance (NMR)
[36] [Reuhs, B.L.; Simsek, S. Nuclear Magnetic Resonance. In Food Analysis, 1st ed.; Nielsen, S.S., Ed.; Food Science Text Series. Springer, Cham, 2017; pp. 151-163. https://doi.org/10.1007/978-3-319-45776-5_10
[37] Truzzi, E.; Marchetti, L.; Benvenuti, S.; Ferroni, A.; Rossi, M.C.; Bertelli, D. Novel strategy for the recognition of adulterant vegetable oils in essential oils commonly used in food industries by applying 13C NMR spectroscopy. J. Agric. Food Chem., 2021, 69(29), 8276-8286. https://doi.org/10.1021/acs.jafc.1c02279
[38] Eisenmann, P.; Ehlers, M.; Weinert, C.H.; Tzvetkova, P.; Silber, M.; Rist, M.J.; Luy, B.; Muhle-Goll, C. Untargeted NMR Spectroscopic Analysis of the Metabolic Variety of New Apple Cultivars. Metabolites 2016, 6(3):29. https://doi.org/10.3390/metabo6030029
Reviewer: Additional info is needed for the table caption, most importantly significant figures.
Authors: As we answered earlier in Major comments, we added additional info in the tables and also changed some of the figure captions as follows:
Figure 4. Time cure test (a) in the range 20–50◦C of native cocoa butter (sample A) and (b) in the range 5-60 °C of modified samples D (in green) compared with that of palm oil (PO, in orange). Vertical lines define the temperature transition region.
Has been replaced by:
Figure 4. Time cure test (a) in the range 20-50◦C of native CB (sample A), of modified (b) sample B, and (c) sample C in the temperature range 15-50°C. (d) Comparison of modified samples D (in green) with that of palm oil (PO, in orange) in the range 20-60 °C. Vertical lines define the tem-perature transition region..
Figure 11. Comparison of 13 C NMR spectra (500 MHz, 298 K, in CDCl3) for natural butter (sam-ple A, in blue) and modified butter (sample B, green) in the range of (a) the α-carbons signals C-2 [33.80 ppm – 34.50 ppm], (b) olefinic signals of C-10 and C-12 of di-unsaturated linoleic acid [127.70 ppm – 128.30 ppm] and (d) C-9 and C-10 of oleic acid and C-13 of linoleic acid [129.50 ppm – 128.30 ppm], (d) of the carbonylic carbon signals of monounsaturated and saturated fatty acids [172.50 ppm – 173.60 ppm].
Has been replaced by:
Figure 11. Comparison of 13 C NMR spectra (125 MHz, 298 K, in CDCl3) for natural CB (sample A, in blue) and modified butter (sample B, green) in the range of (a) the α-carbons signals C-2 [33.80 ppm – 34.50 ppm], (b) olefinic signals of monounsaturated and di-unsaturated acyl chains [127.70 ppm - 130.30 ppm] and (c) of the carbonylic carbon signals of monounsaturated and saturated fatty acids [172.50 ppm – 173.60 ppm].
Reviewer: In the MM section, what is the quality control (QC) data? There is no mention of the QC.
What is the accuracy of the instruments, recovery, LOD, and LOQ ……. These parameters are needed to report the efficiency of any analytical system.
In general, how many times you’ve recorded the data? duplicate? Triplicate?.. what you mentioned in the text is not clear, please elaborate more on this
Auhtors: Thank’s for the comment. As we specified in the manuscript, i.e. from line 435 to line 445 (original manuscript): “ …..13C NMR spectroscopy with inverse gated 1H- decoupling, the signals can be integrated to obtain the relative percentages of the modified TAGs SSU-type in samples B, C and D. This is a relative quantitative determination…….”, then, the integrations of 13C peaks was used to make only a relative ratio between the two types of TAGs, SUS-type and SSU-type, among natural CB and modified samples B, C, and D. To this end, we use the integration of carbonyl carbons (C-1) and the a-carbons (C-2) signals in the position sn-1,3 and sn-2 of the glycerol backbone, considering the assignations reported in figure 11, in table 3 and in the text from line 400 to line 410 (original manuscript). Therefore, our data in table 4 are relative percentage and not an absolute measure, then in our opinion it is not necessary to report the value of LOQ and, on the contrary we report in the text the LOD value and the new reference [38].
To be clearer we added at page 15, after line 441 (original manuscript), after “……were taken into consideration.”
“To note that the lower limits of detection (LOD) are typically with an order of magnitude around micromolar (about 10-6) [38].”
In addition, as specified in the last phrase of the description in Table 4 (“13C NMR results are the mean of three replicates: for each integration was carried out considering the carbonyl (C-1) and α-carbons (C-2) signals)”, each sample was analyzed in triplicate. Also, in the text, page 15, from line 443 to line 445 (original manuscript) we have already reported: “The data of this integration process are reported in Table 4 for samples A, B, C, and D and they are the results of three replicates mean”.
Moreover, currently further and more in-depth studies are being carried out on these kinds of samples and, in the perspective of an absolute quantification, a validation (requiring, others to LOQ and LOD determination, the testing of linearity, robustness, parameters of accuracy, specificity, and selectivity) and quality control (QC) processes are still performing.
Reviewer: R&D section. These sections are repeating information already presented and explain things in an unnecessarily complicated way. The quality of the manuscript would benefit from the whole section being condensed, Line 201-225, Line271-292, Line 297-321, Line 359-424.
Authors: According to the indication of the referee we reviewed all the parts he highlighted as follows:
At pag. 5, from line 201 to line 225 (original manuscript), the sentence: “ Fats in general exhibit a characteristic crystallization behavior which is strongly affected by both the chemical composition of the TAGs and their positional distribu-tion. The crystallization process is commonly studied by Differential Scanning Calo-rimetry (DSC) and these measurements can be more or less accurate depending on the experimental information to be obtained. Indeed, DSC is a routinely performed ther-mal method commonly used to measure the temperature and heat flux associated with the typical transitions in a sample, the enthalpies of fusion (ΔHf), the glass transitions and the crystallization kinetics of materials. For example, the thermal behavior of both natural CB and cocoa butter equivalent (CBE) has been extensively investigated with DSC methods in order to study the different crystalline forms they may present, as demonstrated by the plethora of papers present in the literature [22-28]. In this study, DSC melting profiles were useful and informative tools to make a first and rapid as-sessment of the melting change behavior of our samples as a function of the modifica-tion process. To this end, only the melting profiles generated by calorimetric curves de-rived from the second thermal ramp (DSC profiles obtained from step 5) were taken into consideration and compared. Figure 2 shows these melting profiles for all the samples analyzed. As it can be seen, the melting thermogram of natural CB, sample A, shown in Figure 2.a), is characterized by a rather large endothermic peak centered at about 29°C, indicating a single-phase transition from the solid to the liquid state. It should be noted that since natural CB is a mixture of TAGs, almost exclusively of SUS-type as POS, SOS, and POP, we must talk about melting range instead of single melting temperature, hence the broadened peak in the thermogram. Moreover, the thermogram displayed in Figure 2.a agrees with the DSC data reported in various works [9,23-28] in which the authors, by using much more accurate DSC experiments than ours, associate the melting range to the various crystalline forms that CB can present.”
Has been replaced by:
“The thermal behavior of both natural CB and cocoa butter equivalent (CBE) has been extensively investigated by DSC methods in order to study the different crystalline forms and the melting behavior they may present, as demonstrated by the plethora of papers present in the literature [40,46]. In the present study, the DSC melting profile of the native CB and of the three modified samples (sample B, C and D) has been used to make a first and rapid assessment of their melting temperature as a function of the modification process.”
At pag. 7, from line 271 to line 292 (original manuscript), the sentence: “ In line with calorimetric results, the rheological measurements were carried out for the determination of the mechanical properties and consistency of the materials and to better understand the behavior of the samples seen before. In particular Dynamic temperature ramp (Time cure experiments) test, that allows us to follow the evolution at constant frequency of the viscoelastic modules, G’ (elastic modulus) and G’’ (viscous modulus) against Temperature, were recorder on our samples. Figure 4.a shows the Time cure experiments for natural butter that act as a liquid-like Newtonian system and once again, shows a clear phase transition from solid to liquid around 30°C where a crossover is observed between the two moduli until the complete disappearance of the elastic modulus at 34°C. Instead, among modified fats, the interesting is, once again, sample D, that seems to have improved chemical-physical properties. Its time cure is reported in Figure 4.b compared with that of Palm Oil (PO): as seen previously for the calorimetric behavior, also the rheological behaviors of these two fats are very similar to each other. Sample D shows two different phase transitions: from solid to viscoelastic around 36°C and from viscoelastic to liquid around 50 °C. So, although both samples are made up of mixtures of different TAGs, the native cocoa butter is like a pseudo-single component system, in which statistically the symmetric SUS-type TAGs are prevalent; on the other hand, we could assume that the modified one is a more complicated and stable polymorphic systems with more mixtures of TAGs and where USS-type TAGs cause better packing, strongly increase the transition temperature at which the sample is completely melted and that become significantly higher (also up to about 53-54°C, as in this case).”
It has been completely rephrased as follows:
“In Figure 4., the time cure curve of native CB and modified samples, B, C, and D are reported. Figure 4.a shows the time cure experiments for native CB whose mechanical behavior is that of a liquid-like Newtonian system and shows a clear phase transition, indicated by a sharp change in slope for both moduli (G’ and G’’), from solid to liquid around 30°C. Around this temperature a crossover is observed between the two modu-li until the complete disappearance of the elastic modulus at around 34°C. Instead, the time cure tests of modified samples are more complex. Indeed, for sample C (Figure 4.c) it is still possible to observe a single transition from solid to liquid, but this occurs gradually, with a smooth change in the slope of the two moduli, and in the tempera-ture range from 23 to 40 °C. Instead, samples B and D, Figure 4.b and 4.d, respectively, show a similar profile with two different phase transitions (solid to viscoelastic and viscoelastic to liquid) but with distinct temperature ranges. Sample B is solid until 29 °C, then its behavior is viscoelastic in the temperature range of 29-35 °C and finally, it becomes liquid at 39 °C. Sample D is solid until around 36 °C, then it is viscoelastic between 36°C and 45°C, and becomes liquid at around 50 °C. These experimental re-sults are in line with those obtained from the DSC experiments and corroborate the advanced hypothesis that although all the samples are made up of mixtures of differ-ent TAGs: the native CB, and in part also sample C, appears as a pseu-do-single-component system, in which SUS-type symmetric TAGs (i.e. POP, POS, and SOS) statistically prevail; the modified samples B and D are more complicated systems with more mixtures of TAGs in which some SUS-type TAGs are converted into SSU-type ( i.e. PPO, PSO, SPO, SSO) causing better packing and strongly increasing the transition temperature at which both samples are completely melted (even up to at about 53-54°C). Moreover, it is interesting to note that the rheological behavior of sample D is, once again, very similar to that of PO as can be seen in Figure 4.d where the comparison between their time cure curves is shown.”
At pag. 8, from line 297 to line 321 (original manuscript), the sentence: “At a later time, in order to have a complete overview of the solid-state characteristics of the modified CB fats, powder X-Ray diffraction (PXRD) measurements were also con-ducted on our samples and on native CB and Palm oil. PXRD is the most common tech-nique used to evaluate the crystal packing and the polymorphic state of solid and semi-solid materials, both of which could influence the rheology of the samples [21,22,30]. For the sake of this work, however, a procedure was followed which minimized the possi-bility of random packing, allowing for measurement reproducibility and for a direct, comparative analysis of all samples in the operational experimental conditions, which was our goal at this stage, rather than a deep investigation of the polymorphism eventu-ally showed by each examined fat. In particular, three heating and cooling cycles were performed on each sample before the measurement, and all the samples were manipulat-ed in parallel, placed in the same type of holder, and measured in batteries (one after the other). The result of the conducted study is summarized in figure 5.a-c, in which the PXRD profiles of native CB, modified samples B and D, and Palm oil, are shown and selectively compared. The most interesting aspect of the performed qualitative investigation is that the more modified is the fat (sample D has the highest degree of modification, as it will be later confirmed by NMR data) with SUS-type TAGs converted into SSU-type, the more its PXRD profile is closer to that of Palm oil, in accordance with calorimetric and rheological data (figure 5.c). Another interesting aspect is that while natural CB starts forming grains as a consequence of the multiple heating/cooling cycles, modified sample D doesn’t, as it could be easily observed by eyes, thus resembling Palm oil (figure 5.d). From the data in hand, it seems that a correlation could possibly exist between the formation of grains and the number of peaks in the X-ray diffractogram, which, regardless of the specific peak as-signment to one crystalline phase or another [21,22], could directly correlate the PXRD profile of a fat to a chemical-physical characteristic relevant to the food industry.”
Has been replaced by:
“At a later time, in order to have a complete overview of the solid-state characteristics of the modified CB fats, powder X-Ray diffraction (PXRD) measurements were also con-ducted on our samples and on native CB and PO. PXRD is the most common technique used to evaluate the crystal packing and the polymorphic state of solid and semi-solid materials, both of which could influence the rheology of the samples [39,40,48]. For the sake of this work, however, a procedure was followed that minimized the possibility of random packing, allowing for measurement reproducibility and for direct, comparative analysis of all samples in the operational experimental conditions, which was our goal at this stage, rather than a deep investigation of the polymorphism eventually showed by each examined fat. In particular, three heating and cooling cycles were performed on each sample before the measurement, and all the samples were manipulated in parallel, placed in the same type of holder, and measured in batteries (one after the other). The result of the conducted qualitative investigation is summarized in figure 5.a-c, in which the PXRD profiles of native CB, modified samples B and D, and PO, are shown and selectively com-pared. This study confirmed that the more modified is the fat (sample D has the highest degree of modification, as it will be later confirmed by NMR data) with SUS-type TAGs converted into SSU-type, the more its PXRD profile is closer to that of PO, in accordance with calorimetric and rheological data (Figure 5.c).”
At pag. 11, from line 359 to line 424 (original manuscript), as we think it is unnecessary and may be unclear to the reader, we have decided to remove all the text from line 373 to line 385, i.e. “In particular, signals clearly identifiable in the spectrum of Figure 9 are those relating: a) to the carbonyl groups (C-1); b) to the alpha (C-2) and beta (C-3) carbon atoms to the carbonyl group; c) to the final methyl groups ω-1 (C-16 of palmitic acid, C-18 of stearic, oleic and linoleic acid); d) to the methylene groups ω-2 (C-17 of stearic, oleic and linoleic acid and C-15 of palmitic acid); e) to the methylene groups ω-3 (C-16 of stearic, oleic and linoleic acid and C-14 of palmitic acid); f) to the carbon atoms of double bonds ( C-9, C-10, C-12 and C-13 of linoleic acid and C-9 and C-10 of oleic acid). The rest of the carbons along the chain (generic methylene groups -CH2) which are further away from the extremities or from the unsaturation sites, appear very close to each other in a spectral region between 29.00 and 30.00 ppm. Table 3 shows these attributions for all the fatty acids present in natural sample A and the comparison with the corresponding chemical shifts values of the FAs carbons in sample B”.
We have, however, left the rest of the subsection as it is because we think that all the details are needed to arrive at the conclusions outlined.
Reviewer: Line 46-47: A reference is needed here, for example, you can use https://onlinelibrary.wiley.com/doi/book/10.1002/9781118788745
Authors: we followed the reviewer's suggestion and added the references below:
[5] Wijewardene, U.; Premlal Ranjith, H. M. Food safety and quality issues of dairy fats. In Fats in Food Technology, 2nd ed.; Kanes K. Rajah Ed.; John Wiley & Sons, 2014; pp. 289-325. https://doi.org/10.1002/9781118788745.ch8
[6] Abd-Aziz, S.; Gozan, M.; Ibrahim, M.F.; Phang, L.-Y. Demand and Sustainability of Palm Oil Plantation. In Biorefinery of Oil Producing Plants for Value-Added Products, 1st ed.; Abd-Aziz, S., Gozan, M., Ibrahim, M.F., Phang, L.-Y. Eds.; Wiley Online Library, 2022. https://doi.org/10.1002/9783527830756.ch2
[7] Fitzherbert, E. B.; Struebig, M. J.; Morel, A.; Danielsen, F.; Brühl, C. A.; Donald, P. F.; Phalan, B. How will oil palm expansion affect biodiversity? Trends Ecol. Evol. 2008, 10, 538-545. https://doi.org/10.1016/j.tree.2008.06.012
Reviewer Line 371-385: A reference is needed here, for example, you can use: https://doi.org/10.1007/s12161-021-02203-0
Authors: we followed the reviewer's suggestion and we added the reference:
[55] Hama, J.R.; Fitzsimmons-Thoss, V. Determination of Unsaturated Fatty Acids Composition in Walnut (Juglans regia L.) Oil Using NMR Spectroscopy. Food Anal. Methods, 2022, 15, 1226-1236. https://doi.org/10.1007/s12161-021-02203-0]
Reviewer: Figure 11. There is a typo error, (d) repeated5t two times.
Authors: We have corrected the error and also replaced images 11b and c with a single 11.b.
Moreover, the caption of Figure 11 ( original manuscript): “Figure 11. Comparison of 13 C NMR spectra (500 MHz, 298 K, in CDCl3) for natural butter (sample A, in blue) and modified butter (sample B, green) in the range of (a) the α-carbons signals C-2 [33.80 ppm – 34.50 ppm], (b) olefinic signals of C-10 and C-12 of di-unsaturated linoleic acid [127.70 ppm – 128.30 ppm] and (d) C-9 and C-10 of oleic acid and C-13 of linoleic acid [129.50 ppm – 128.30 ppm], (d) of the carbonylic carbon signals of monounsaturated and saturated fatty acids [172.50 ppm – 173.60 ppm].”
Has been replaced by:
“Figure 11. Comparison of 13C NMR spectra (125 MHz, 298 K, in CDCl3) for natural CB (sample A, in blue) and modified butter (sample B, green) in the range of (a) the α-carbons signals C-2 [33.80 ppm – 34.50 ppm], (b) olefinic signals of monounsaturated and di-unsaturated acyl chains [127.70 ppm - 130.30 ppm] and (c) of the carbonylic carbon signals of monounsaturated and saturated fatty acids [172.50 ppm – 173.60 ppm].”

Reviewer 2 Report
In this work, the authors seek to characterize the physico-chemical properties and the triacylglycerol composition of cocoa butter and its derivatives by experiments in NMR, DSC, X-Ray, rheology. In the materials part it is meant that 4 samples will be studied: cocoa butter and 3 other samples which correspond to different modifications of the triacylglycerol composing the cocoa butter without giving us with precision the process leading to these modifications or the new compositions in triacylglycerols of these new samples of cocoa butter (for reasons of secrecy).
An experimental protocol is proposed for the study in DSC which is moreover different from that proposed in X-Rays. Cocoa butter exists in different polymorphic forms. The isolation protocol for these forms is well known. The choice was made to highlight forms 3 and 4 of cocoa butter by DSC, while the one chosen for the X-Rays highlights : cooling from 70°C to room temperature without controlling the cooling slope and then storage at 4°C for 72 hours. Storing the cocoa butter at 4¨C will promote the onset of crystallization in form 5 then in form 6. So between the DSC and X-Rays experiments we do not have the same polymorphic forms. Moreover, this is reflected in the results: in DSC the authors measure a temperature at 29°C without it being known whether it is the Tonset temperature, or Tmax or Toffset which corresponds respectively to the beginning, the top and the end of thermal events. This temperature is closer to a form 4 of cocoa butter than a form 5. On the X-Rays results, we very clearly find the form 5 of cocoa butter from the WAXS results, even if it missing from these results is the SAXS (Small Angle X-Ray Scattering) study, which also gives valuable information on the polymorphic form of the triacylglycerols that make up cocoa butter. So from these two types of DSC and X-Rays experiments we cannot compare the same phenomena. These studies are therefore not complementary. Regarding the results of samples B, C and D, whether in X-Rays or DSC, no interpretation is given because we were not given the modifications of the types of triacylglycerols. This is greatly lacking in the exploitation of the results. The authors could not have given the composition of the cocoa butters but they could have sought to highlight at least one of the elements which would explain these changes. In the end, they only use the NMR results to show us the internal change of SUS-type triacylglycerols to SSU-type. Only in the conclusion are we given an example of triacylglycerols that might be suitable for PPO, PSO and SSO change. To explain the results in X-Rays and DSC it would be necessary to take those existing for these three triacylglycerols and use these results to explain the transformations in X-Rays and DSC knowing that the sample preparation protocol has a direct consequence of these results.
In conclusion, I ask to review the results in DSC and X-Rays by providing details on the exploitation of the curves and by proceeding with the same protocol between the DSC and X-Rays in order to be able to compare the results before giving a favorable opinion to publication.
Author Response
Rende, January 10, 2023
Dear reviewer,
thank you very much for the effort that you spent on our manuscript. We appreciate the very constructive and helpful comments and suggestions to improve our work. In response to your input, we modified the text's recommended modifications.
We hope that our revised manuscript addresses all concerns satisfactorily.
All changes are presented in detail below. The issues raised by the reviewer are set in italics and our answers are in plain font. All our changes are included in the revised manuscript in red color.
Comments and Suggestions for Authors
In this work, the authors seek to characterize the physico-chemical properties and the triacylglycerol composition of cocoa butter and its derivatives by experiments in NMR, DSC, X-Ray, rheology. In the materials part it is meant that 4 samples will be studied: cocoa butter and 3 other samples which correspond to different modifications of the triacylglycerol composing the cocoa butter without giving us with precision the process leading to these modifications or the new compositions in triacylglycerols of these new samples of cocoa butter (for reasons of secrecy).
An experimental protocol is proposed for the study in DSC which is moreover different from that proposed in X-Rays. Cocoa butter exists in different polymorphic forms. The isolation protocol for these forms is well known. The choice was made to highlight forms 3 and 4 of cocoa butter by DSC, while the one chosen for the X-Rays highlights : cooling from 70°C to room temperature without controlling the cooling slope and then storage at 4°C for 72 hours. Storing the cocoa butter at 4¨C will promote the onset of crystallization in form 5 then in form 6. So between the DSC and X-Rays experiments we do not have the same polymorphic forms. Moreover, this is reflected in the results: in DSC the authors measure a temperature at 29°C without it being known whether it is the Tonset temperature, or Tmax or Toffset which corresponds respectively to the beginning, the top and the end of thermal events. This temperature is closer to a form 4 of cocoa butter than a form 5. On the X-Rays results, we very clearly find the form 5 of cocoa butter from the WAXS results, even if it missing from these results is the SAXS (Small Angle X-Ray Scattering) study, which also gives valuable information on the polymorphic form of the triacylglycerols that make up cocoa butter. So from these two types of DSC and X-Rays experiments we cannot compare the same phenomena. These studies are therefore not complementary. Regarding the results of samples B, C and D, whether in X-Rays or DSC, no interpretation is given because we were not given the modifications of the types of triacylglycerols. This is greatly lacking in the exploitation of the results. The authors could not have given the composition of the cocoa butters but they could have sought to highlight at least one of the elements which would explain these changes. In the end, they only use the NMR results to show us the internal change of SUS-type triacylglycerols to SSU-type. Only in the conclusion are we given an example of triacylglycerols that might be suitable for PPO, PSO and SSO change. To explain the results in X-Rays and DSC it would be necessary to take those existing for these three triacylglycerols and use these results to explain the transformations in X-Rays and DSC knowing that the sample preparation protocol has a direct consequence of these results.
In conclusion, I ask to review the results in DSC and X-Rays by providing details on the exploitation of the curves and by proceeding with the same protocol between the DSC and X-Rays in order to be able to compare the results before giving a favorable opinion to publication.
Authors: Thank you for the comments and we would like to elucidate the issues raised by the reviewer.
- We start to clarify the first point raised by the review: “ An experimental protocol is proposed for the study in DSC which is moreover different from that proposed in X-Rays. Cocoa butter exists in different polymorphic forms………….”
We are well aware that cocoa butter exists in various polymorphic forms and the referee's comment would be appropriate if the aim of this work had been the polymorphic study of cocoa butter and its derivatives. Obviously in this case what was said by the referee (i.e. "An experimental protocol is proposed for the study in DSC which is however different from the one proposed in X-Rays"), the experimental protocol to be used in all the techniques necessary for their identification, not only in the DSC and X-Ray experiments, had to be the same and it was necessary to have accurate control over all the various phases performed. Actually, the identification and study of these different crystalline forms is not the goal of our manuscript and we do not want to propose any protocol to determine the various crystalline forms that the analyzed materials can present. The DSC, X-Ray and rheological experiments helped us to identify the differences in the behavior of the analyzed materials, which obviously reflect the differences in their chemical-physical properties. Within each technique we used the same procedure to treat each sample precisely in order to highlight the differences. The evident differences, in terms of both behavioral profiles and melting temperatures, of the three modified fats compared to the initial cocoa butter sample, coming from the different techniques used which agree with each other even though each technique treats the samples differently, allowed us to hypothesize that the modification reaction can lead to the formation of TAGs of the SSU-type which coexist with the SUS-type. This hypothesis was subsequently demonstrated through NMR measurements.
This point is probably unclear as it should be and we would like to elucidate it.
Then, the main objective of this study was to investigate the variations in chemical-physical properties of three fats derived from a sample of natural cocoa butter subjected to an unconventional process, i.e. carried out without enzymes or chemical additives, and to demonstrate experimentally how these variations, in particular the high melting temperatures up to about 50°C, can be explained and rationalized, through NMR measurements, in terms of exclusive manipulation of the positional distribution of the fatty acids leaving their nature unaltered, i.e. following the modification process it has been possible to demonstrate, by analysis of 13C NMR spectra, that positional isomers were obtained in which part of the FAs in the mixture of SUS-type TAGs (mainly POP, POS and SOS) were converted into SSU-type TAGs (mainly PPO, PSO and SSO). Moreover, it was also possible to demonstrate, again from the 13C NMR spectra, that the clear differences (in DSC and PXRD profiles and in the rheological measurements) between the three modified fat samples can be attributed to the different percentages of SSU-type TAGs formation.
In order to clarify this point and make it more obvious to the reader, we have rephrasing and added more details to the last part of the introduction section.
Then, Page 3, line 99 (original manuscript), the sentence in the introduction: “In this work, an experimental investigation has been carried out on the chemical-physical properties of both natural CB and some modified butters obtained from it thought a process acting on the FAs position, but not on their nature, within the naturally present TAGs.
The TAG composition, crystallization, thermal behavior, and rheological properties have been investigated on all the materials to better understand at molecular and macro-scopic level their characteristics and properties. The experimental investigation carried out include: i) DSC tests to examine thermal features; ii) rheological testing to explore mechanical properties; iii) powder X-Ray diffraction to qualitatively compare the obtained matrices at the solid-state level; iv) 1H and 13C NMR (1D and 2D) spectroscopy for a molecular characterization. The accordance between experimental results from these different techniques confirms how a specific chemical and physical behavior is strongly deter-mined by the stereochemistry of TAGs. Particularly NMR allows a qualitative and quantitative (in percentage of TAGs) characterization of these complex mixtures and provides useful information on the positional distribution of FAs”.
Has been replaced by:
“Thus, in this context, the present study was designed to experimentally investigate the changes in the chemical-physical properties of some modified fats derived from a natural CB sample through an unconventional process acting on it and to demonstrate that this modification process has led to a manipulation of the positional distribution of the FAs but has left their nature unaltered, i.e. following the modification process, positional isomers have been obtained in which part of the FAs in the mixture of SUS-type TAGs (mainly POP, POS and SOS) were converted into SSU-type TAGs (mainly PPO (1,2-palmitoyl-3-oleoyl-sn-glycerol), PSO (1-palmitoyl-2- stearyl -3-Oleoyl-sn-glycerol) and SSO (1,2-stearyl-3-oleoyl-sn-glycerol)).
To this end, a comparative study among the native CB sample, three samples of fats derived from it and a sample of PO was carried out. The TAG composition, crystallization, thermal behavior and rheological properties have been studied on all these materials in order to understand their characteristics, properties and differences at the molecular and macroscopic level. In particular, the experimental investigation carried out included: i) Differential Scanning Calorimetry (DSC) tests to examine thermal features (principally melting points); ii) rheological testing to explore mechanical properties; iii) powder X-Ray diffraction (PXRD) to qualitatively compare the obtained matrices at the solid-state level; iv) 1H and 13C Nuclear Magnetic Resonance (NMR, 1D, and 2D) spectroscopy for molecular characterization. It should be emphasized that the NMR technique was particularly useful in the qualitative and quantitative (in the percentage of TAGs) characterization of these complex mixtures and, basically, it allowed to experimentally confirm the positional rearrangement of the acyl chains that occurred during the modification process.
The accordance between experimental results from these different techniques confirms how a specific chemical and physical behavior is strongly determined by the stereochemistry of TAGs.”
- Authors: We would like to underline that investigations are still ongoing in order to understand and correctly identify the various crystalline forms of the three modified fats and that the experiments reported in this manuscript are purely qualitative as we have underlined in various parts of the work.
For example:
- At page 5, line 211 (original manuscript) we reported: “In this study, DSC melting profiles were useful and informative tools to make a first and rapid assessment of the melting change behavior of our samples as a function of the modification process”.
- At page 8, line 301 (original manuscript) we reported: “For the sake of this work, however, a procedure was followed which minimized the possibility of random packing, allowing for measurement reproducibility and for a direct, comparative analysis of all samples in the operational experimental conditions, which was our goal at this stage, rather than a deep investigation of the polymorphism eventually showed by each examined fat.”
- At page 8, line 310 (original manuscript) we reported: “The most interesting aspect of the performed qualitative investigation is that the more modified is the fat (sample D has the highest degree of modification, as it will be later con-firmed by NMR data) with SUS-type TAGs converted into SSU-type, the more its PXRD profile is closer to that of Palm oil, in accordance with calorimetric and rheological data (figure 5.c).”
- Reviewer: These studies are therefore not complementary. Regarding the results of samples B, C and D, whether in X-Rays or DSC, no interpretation is given because we were not given the modifications of the types of triacylglycerols. This is greatly lacking in the exploitation of the results. The authors could not have given the composition of the cocoa butters but they could have sought to highlight at least one of the elements which would explain these changes. In the end, they only use the NMR results to show us the internal change of SUS-type triacylglycerols to SSU-type. Only in the conclusion are we given an example of triacylglycerols that might be suitable for PPO, PSO and SSO change. To explain the results in X-Rays and DSC it would be necessary to take those existing for these three triacylglycerols and use these results to explain the transformations in X-Rays and DSC knowing that the sample preparation protocol has a direct consequence of these results.
Authors: As we have already underlined previously, the DSC measurements and the powder X-Ray diffraction measurements were qualitative measurements and did not serve for a deep investigation of the polymorphism shown by each fat examined. On the other hand, however, they served us to highlight the differences between each sample and above all the differences in the melting temperatures which, being very high (up to 54 °C for sample D), as the review well knows, cannot be explained with different crystalline forms in which the same types of TAGs (SUS-type) can be packed.
The review said that: “This is greatly lacking in the exploitation of the results. The authors could not have given the composition of the cocoa butters but they could have sought to highlight at least one of the elements which would explain these changes”, but we believe we have indicated for each investigation technique used the elements that led us to hypothesize a change in the rearrangement of the positions occupied by the various fatty acids on the glycerol skeleton.
For example:
- At Page 5, line 230 (original manuscript), we reported: “These data clearly indicate that the modified butter samples present a much more complicated polymorphic system, with high transition temperatures (up to about 54°C), which is difficult to explain because of the presence of different crystalline forms due to the unique TAG mixture present in the natural butter. To attempt rationalize this effect we hypothesized that, as it will be confirmed later by the molecular characterization via-NMR, the modification reaction may lead to the formation of SSU-type TAGs which coexist with the SUS-type. In this case mixtures of TAGs with different stereochemistry in different ratios could have a more or less strong effect on the melting point.”
- At Page 7, line 286 (original manuscript), we reported: “ So, although both samples are made up of mixtures of different TAGs, the native cocoa butter is like a pseudo-single component system, in which statistically the symmetric SUS-type TAGs are prevalent; on the other hand, we could assume that the modified one is a more complicated and stable polymorphic systems with more mixtures of TAGs and where SSU-type TAGs cause better packing, strongly increase the transition temperature at which the sample is completely melted and that become significantly higher (also up to about 53-54°C, as in this case)”.
- At Page 8, line 310 (original manuscript), we reported: “The most interesting aspect of the performed qualitative investigation is that the more modified is the fat (sample D has the highest degree of modification, as it will be later con-firmed by NMR data) with SUS-type TAGs converted into SSU-type, the more its PXRD profile is closer to that of Palm oil, in accordance with calorimetric and rheological data (figure 5.c).”
In addition, these measurements, including also the rheological experiments, have served to show, at a qualitative level, that the behavior of sample D (i.e. the modified fat with the highest degree of modification) is similar to that of palm oil.
- Reviewer: In conclusion, I ask to review the results in DSC and X-Rays by providing details on the exploitation of the curves and by proceeding with the same protocol between the DSC and X-Rays in order to be able to compare the results before giving a favorable opinion to publication.
Authors: As we answered previously, the main purpose of the present study was to experimentally investigate the changes in the chemical-physical properties of the analyzed modified fats and to relate them to the different conditions of the modification process by demonstrating that this treatment led to a manipulation of the positional distribution of the FAs but left their nature unchanged. Therefore, an in-depth investigation of the polymorphism possibly highlighted by each fat examined by DSC and X-ray measurements has not been contemplated in this work. The melting thermograms and the PXRD patterns reported in the manuscript are only qualitative.
However, other studies are still in progress aimed at investigating the possible crystalline forms presented by these three modified samples (samples B, C and D) that include also accurate DSC measurements and other appropriate experiments. Anyhow, we followed the reviewer's advice, and we repeated the DSC measurements treating the samples with the same procedure (protocol) used to obtain the PXRD patterns, i.e. samples were dissolved at a temperature of ca. 70 °C, transferred in the sample holder, slowly cooled down to room temperature and stored at 4°C for 24 hours, twice. Lastly, dissolved again at ca. 70 °C, cooled down to room temperature and stored at 4°C for other 72 hours before the measurements. After this preparation the samples were treated with the same procedure of heating/cooling cycles as reported at subsection 2.2, from line 130 to line 135 (original manuscript).
The following figure shows the melting thermograms of the native CB (Figure 1) and of the sample D (Figure 2) obtained from the procedure above descripted in two different steps of the DSC cycles. Indeed, the graphs (a) in both figures are referred to the first heating process (step. 2 of the heating/cooling sequence), while the two graphs (b) are referred to the second heating process (step. 5 of the heating/cooling sequence). As can be seen from the two figures, the graphs (b) can be compared to the same reported in the manuscript for both samples, the only differences are in the transition temperatures that are slightly different from what reported in the manuscript. We would like to emphasize that the interesting thing concerns sample D, which has a behavior in terms of melting temperatures similar to that already reported in the manuscript. Since investigations into the crystalline forms of modified fats are still ongoing, we have decided not to report these diagrams in the manuscript, and we would like to ask the reviewer to keep these thermograms to himself without disclosing them.
|
|
|
|
(a) |
(b) |
Figure 1. Melting profile (endo-down) of native CB recorded during (a) the first heating process (step. 2 of the heating/cooling sequence described at line 132 of the section “2.2 Differential Scanning Calorimetry”, original manuscript) of the DSC experiments from 15 °C to 70 °C at 2 °C/min (yellow curve) and (b) heating process (step. 5 of the heating/cooling sequence described at line 135 of the section “2.2 Differential Scanning Calorimetry”, original manuscript) of the DSC experiments from 15 °C to 70 °C at 2 °C/min (orange curve).
|
|
|
|
(a) |
(b) |
Figure 2. Melting profile (endo-down) of sample D recorded during (a) the first heating process (step. 2 of the heating/cooling sequence described at line 132 of the section “2.2 Differential Scanning Calorimetry”, original manuscript) of the DSC experiments from 15 °C to 70 °C at 2 °C/min (pink curve) and (b) heating process (step. 5 of the heating/cooling sequence described at line 135 of the section “2.2 Differential Scanning Calorimetry”, original manuscript) of the DSC experiments from 15 °C to 70 °C at 2 °C/min (red curve).

Round 2
Reviewer 1 Report
The authors addressed all of my comments for the manuscript. The revised manuscript improved substantially.
Therefore, I recommend the manuscript to be accepted and published.
Best wishes,
Reviewer 2 Report
I appreciate the effort of the authors to respond to the remarks I had given. The answers to my questions are very satisfactory. I recommend the article for publication.